# Monospecific and bispecific monoclonal SARS-CoV-2 neutralizing antibodies that maintain potency against B.1.617

Lei Peng[1,2,3,20], Yingxia Hu[4,20], Madeleine C. Mankowski[5,6,20], Ping Ren[1,2,3,20], Rita E. Chen[7], Jin Wei[5,6], Min Zhao[8], Tongqing Li [9,10], Therese Tripler[4], Lupeng Ye [1,2,3], Ryan D. Chow [1,2,3,11,12], Zhenhao Fang[1,2,3], Chunxiang Wu [4], Matthew B. Dong[1,2,3,6,11,13], Matthew Cook[4], Guilin Wang[14], Paul Clark[1,2,3], Bryce Nelson[9,10], Daryl Klein [9,10], Richard Sutton[8], Michael S. Diamond [7,15], Craig B. Wilen [5,6,21✉], Yong Xiong [4,21✉] & Sidi Chen [1,2,3,12,13,16,17,18,19,21✉]

COVID-19 pathogen SARS-CoV-2 has infected hundreds of millions and caused over 5 million deaths to date. Although multiple vaccines are available, breakthrough infections occur especially by emerging variants. Effective therapeutic options such as monoclonal antibodies (mAbs) are still critical. Here, we report the development, cryo-EM structures, and functional analyses of mAbs that potently neutralize SARS-CoV-2 variants of concern. By high-throughput single cell sequencing of B cells from spike receptor binding domain (RBD) immunized animals, we identify two highly potent SARS-CoV-2 neutralizing mAb clones that have single-digit nanomolar affinity and low-picomolar avidity, and generate a bispecific antibody. Lead antibodies show strong inhibitory activity against historical SARS-CoV-2 and several emerging variants of concern. We solve several cryo-EM structures at ~3 Å resolution of these neutralizing antibodies in complex with prefusion spike trimer ectodomain, and reveal distinct epitopes, binding patterns, and conformations. The lead clones also show potent efficacy in vivo against authentic SARS-CoV-2 in both prophylactic and therapeutic settings. We also generate and characterize a humanized antibody to facilitate translation and drug development. The humanized clone also has strong potency against both the original virus and the B.1.617.2 Delta variant. These mAbs expand the repertoire of therapeutics against SARS-CoV-2 and emerging variants.

---

A full list of author affiliations appears at the end of the paper.

In the ongoing coronavirus disease 2019 (COVID-19) pandemic, severe acute respiratory syndrome coronavirus (SARS-CoV-2) has infected over 270 million individuals, resulting in more than 5 million deaths around the globe[1]. Although multiple vaccines are available, breakthrough infections still occur[2], especially with variant strains. Thus, broadly effective therapeutic options are critical for medical treatment. Monoclonal antibodies (mAbs) have been effectively deployed for the prevention or treatment of COVID-19[3]. However, the emergence of mutations in spike[4] and new variant lineages calls for developing additional therapeutic interventions, including mAbs with potent and broadly neutralizing ability.

Certain variants affect the rate of spread and/or even the ability to evade immune recognition, potentially dampening the efficacy of antibody therapy or vaccines, and have been designated by WHO and CDC as "variants of concern" (VoC)[3]. The B.1.1.7 lineage (Alpha variant) has an increased rate of transmission and higher mortality[5]. The B.1.351 lineage (Beta variant) has an increased rate of transmission, resistance to antibody therapeutics, and reduced vaccine efficacy[6–8]. The lineage B.1.617 including B.1.617.1 (Kappa variant), B.1.617.2 (Delta variant), and B.1.617.3 have emerged and become dominant in multiple regions in the world[9,10]. The B.1.617 lineage has an increased rate of transmission, shows reduced-serum antibody reactivity in vaccinated individuals, and exhibits resistance to antibody therapeutics under emergency use authorization (EUA)[11–15]. The Delta variant has become dominant in the United States and many countries across the globe[16–18]. A newly emerged variant Omicron (B.1.1.529) with extensive mutations in the Spike gene also shows rapid transmission[19–21]. With these VOCs, even emergency-use authorized (EUA) mAb therapies have faced challenges with resistant viral variants, causing some to be withdrawn, highlighting a need for more mAb candidates in development. The Discovery of neutralizing antibodies with broad neutralizing activities or multi-specific antibodies, which can increase antibody efficacy and prevent viral escape, might improve the countermeasure arsenal against COVID-19[3].

The majority of preclinical and clinical SARS-CoV-2 mAbs were discovered utilizing the blood of COVID-19 patients[3]. In comparison, immunization of mice followed by hybridoma screening is a standard method for discovering therapeutic mAbs against viruses[22]. However, hybridoma screening for potent neutralizing mAbs from immunized mice is a slow and laborious process. The recent development of high-throughput single-cell technologies enabled direct sequencing of fully recombined VDJ sequences of B cell receptor (BCR) repertoires from single cells[23,24]. This technology has successfully been used to isolate human neutralizing mAbs against pathogens such as HIV, Ebola viruses, and recently SARS-CoV-2[25–29].

The SARS-CoV-2 surface spike glycoprotein (S) mediates entry into target cells and is a primary target of neutralizing antibodies[30–32]. SARS-CoV-2 spike is a trimer in the pre-fusion form, consisting of three copies of S1 and S2 subunits[33,34]. The S1 subunit is composed of an N-terminal domain (NTD) and a C-terminal receptor-binding domain (RBD) that recognizes the host angiotensin-converting enzyme 2 (ACE2) receptor on the cell surface[31,35–37]. The S2 subunit contains the fusion peptide, along with other key regions, to induce membrane fusion of the virus and the target cell[38]. Spike RBD is flexible, moving between "up" and "down" conformations but only binds to ACE2 in the up conformation[34,36,39]. Previous studies have suggested that most of the potent SARS-CoV-2 neutralizing antibodies target spike RBD through an interface directly overlapping with the ACE2-binding surface[40,41].

Here, we report the rapid identification of highly potent SARS-CoV-2-neutralizing mAbs using high-throughput single-cell BCR

sequencing from mice immunized with purified SARS-CoV-2 RBD. We generate a bispecific antibody with two antigen-recognition variable regions from two of the top mAb clones against SARS-CoV-2 RBD. These monospecific and bispecific antibodies display high affinity and avidity to the RBD-antigen, and potently neutralized historical and B.1.617 lineage viruses. We also generate and characterize a humanized antibody to facilitate clinical development. This clone shows strong potency against both the original virus and the Delta variant, both in vitro and in vivo. We resolve the three-dimensional cryo-EM structures of these neutralizing antibodies in complex with the spike-ectodomain trimer, which show the epitopes, as well as the combinations of open and close RBD conformations of trimeric spike, bound to the mAbs. Structure-based mutation and epitope analysis reveal the neutralization potency of the lead mAb clones against B.1.617 variants.

## Results

**Single-cell BCR sequencing of RBD-immunized mice identified enriched BCRs encoding strong mAbs against SARS-CoV-2.** To generate potent and specific mAbs against SARS-CoV-2, we immunized two different mouse strains: C57BL/6 J and BALB/c with RBD protein with a C-terminal hexahistidine tag following a standard 28-day immunized protocol (Supplementary Fig. 1a). Using anti-mouse CD138 beads, we isolated progenitor B cells and plasma B cells from spleen, lymph node, and bone marrow of selected immunized mice that showed high serum binding titers against RBD (Supplementary Fig. 1a). We performed single-cell VDJ (scVDJ) sequencing on the isolated B cells (Supplementary Fig. 1a). The scVDJ data revealed the landscape of immunoglobulin clonotypes in immunized mice (Fig. 1a and Supplementary Dataset 1) and identified enriched IgG1 clones (Fig. 1b and Supplementary Dataset 1). We took the VDJ sequences from 11 top-ranked clones by clonal frequency. We then cloned the paired variable region of the heavy and light chain into human IgG1 (hIgG1) heavy and light chain backbone vectors separately, for antibody reconstruction utilizing the Expi293F mammalian expression system.

**Anti-Spike RBD monoclonal antibodies have single-digit nanomolar affinity and low-picomolar avidity.** After expression and purification of hIgG1 antibody clones, we tested their reactivity against SARS-CoV-2 spike RBD by ELISA. Eight of the eleven mAbs showed positive RBD-binding (Supplementary Fig. 1c). We then screened mAb clones showing a high RBD ELISA positive rate for their neutralizing ability using an HIV-1-based SARS-CoV-2 pseudovirus system and a VSV-based SARS-CoV-2 pseudovirus system (Supplementary Fig. 2a–c). Two mAbs, Clones 2 and 6, showed the strongest binding and neutralizing activity (Supplementary Fig. 2a–c). As SARS-CoV-2 continues to mutate and evolve, leading to variants, it is critical to prevent viral escape from antibody recognition. To overcome this problem, antibody combinations from two or more mAbs have been developed and utilized[42]. As an alternative, a single bispecific mAb can be used, as it can recognize two epitopes. One advantage of bispecific mAbs is that a single antibody product can be manufactured instead of two separate mAbs, which in theory could reduce the cost and formulation complexity. Thus, we generated a bispecific antibody using the antigen-specific variable regions of both Clones 2 and 6 (named as Clone 16). To generate the bispecific antibody, we utilized a "knobs into holes" (KiH)-CrossMab methodology, and to ensure the correct heterodimerization of the two different heavy chains and correct pairing of heavy and light chains of each variable region (Supplementary Fig. 1d). We validated the antigen-specificity of the bispecific

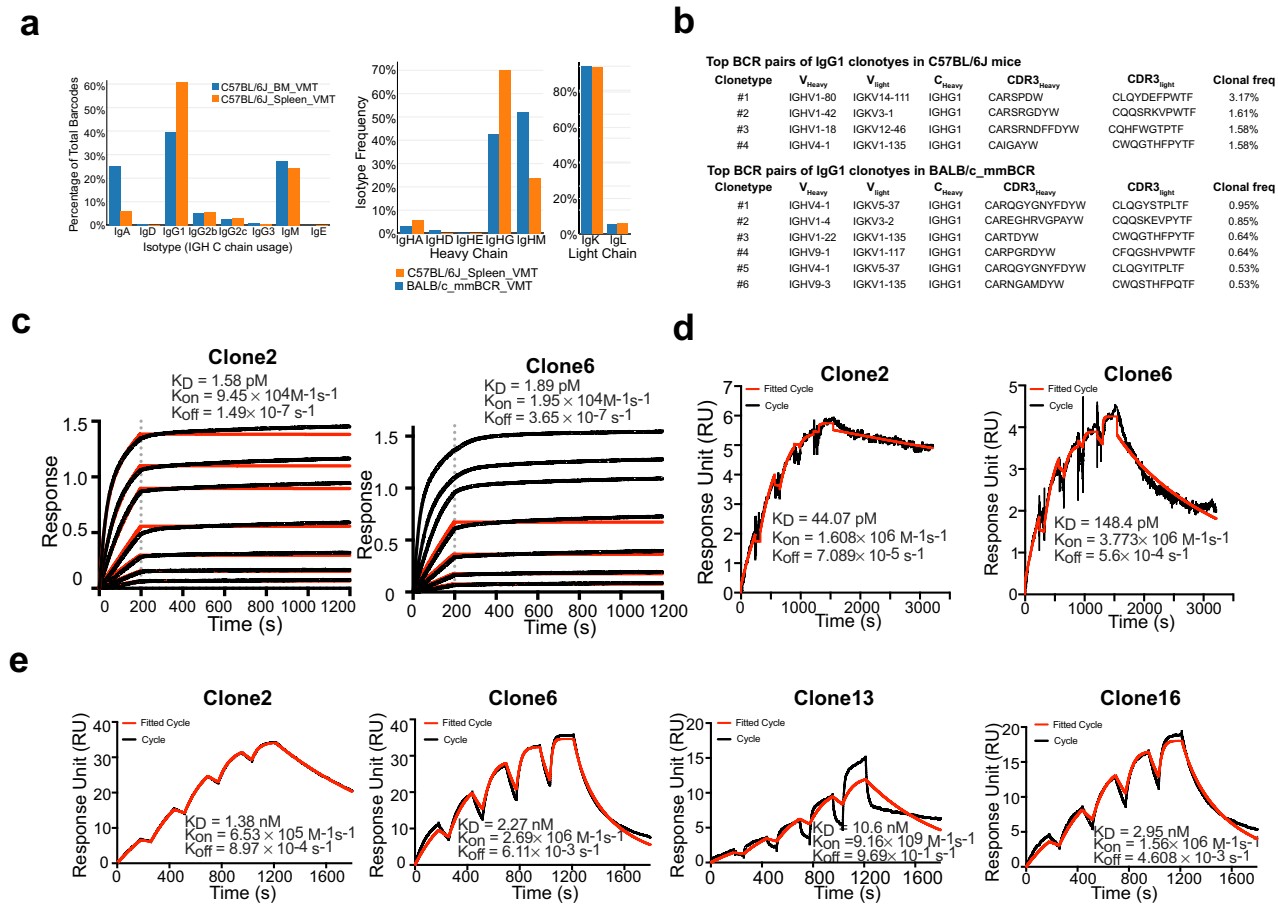

**Fig. 1 Immunization-single B cell sequencing leads to development of potent monoclonal antibodies in both monospecific and bispecific forms.**
**a** Frequency distribution of immunoglobulin isotypes in SARS-CoV-2 Spike RBD (receptor-binding domain) immunized mice. (Top) Analysis of isotypes in both bone marrow and spleen in RBD-immunized C57BL/6 J mice ($n = 1$ mouse) (Bottom) Analysis of isotypes in RBD-immunized C57BL/6 J and BALB/c mice ($n = 1$ mouse). **b** CDR3 sequences of heavy and light chains of the top enriched antibody clones from RBD-his tag immunized C57BL/6 J mice (Top) and BALB/c mice (Bottom). CDR3: Complementarity-determining regions 3. **c** Octet measurement of binding strengths top SARS-CoV-2 RBD-specific monospecific mAb clones (Clones 2 and 6). The binding was particularly strong, thus that the dissociation stage was never observed in this BLI assay. **d** SPR measurement of binding strengths top SARS-CoV-2 RBD-specific monospecific mAb clones (Clones 2 and 6), using an NTA chip where the antigen (RBD) was fixed on the chip. **e** SPR measurement of binding strengths top SARS-CoV-2 RBD-specific monospecific mAb clones (Clones 2 and 6), a humanized mAb clone (Clone 13), and a bispecific mAb (Clone 16), using an Fc chip where antibodies were fixed on the chip. Source data and additional statistics for experiments are provided as a Source Data file.

antibody along with its two parent mAbs using ELISA (Supplementary Fig. 1e).

To characterize the biophysical nature of the RBD reactivity of lead clones, we performed biolayer interferometry (BLI) and surface plasmon resonance (SPR) (Fig. 1c–e). BLI results showed that Clones 2 and 6 bound to the RBD with picomolar level dissociation constant (Kd) (Fig. 1f). The binding is particularly strong so that the dissociation stage was never observed in the BLI assay (Fig. 1c). SPR using an NTA-CHIP based assay with the RBD-antigen immobilized also revealed picomolar level dissociation constant (Kd) for Clones 2 and 6 (Fig. 1d). In the NTA-CHIP, where the RBD-antigen is immobilized, multiple binding events can occur, leading to potential trimer formation of RBD protein and measurement of avidity. In parallel, we performed SPR with a CM5 CHIP-based assay with pre-coated antibodies, which measures monovalent binding between RBD and the antibody. Results from the CM5 CHIP SPR assay revealed a strong affinity with single-digit nanomolar Kd of Clone 2 and Clone 6 (Fig. 1e). A humanized Clone 2 (named Clone 13) maintained a strong binding affinity to RBD at low-double-digit nanomolar Kd (Fig. 1e). The bispecific Clone 16 also displayed

high affinity to RBD also at single-digit nanomolar Kd (Fig. 1e), although not significantly higher than either parental clone alone.

**Cryo-EM structures of lead antibody clones bound to the SARS-CoV-2 spike define epitopes and binding conformations.**
We then generated Fab fragments of the lead clones and determined the cryo-EM structures of Clone 2 or 6 Fabs in complex with the ectodomain of SARS-CoV-2 spike trimer (S trimer) at ~3 Å resolution (Table 1). In both cases, an S trimer is bound with three Fab molecules, one per RBD in various conformations. In total, we determined five different Fab-S trimer complex structures, each with the S trimer in a specific conformational state. Clone 2 Fab binds to S trimer in two states, one with 2 RBDs in the up conformation (60% of all complexes) and one with all 3 RBDs up (40%) (Fig. 2a and Supplementary Fig. 3a). Clone 6 Fab binds to S trimer in three states, one with all three RBDs down (26% of all complexes), one with 1 RBD up (43%) and one with two RBDs up (31%) (Fig. 2b and Supplementary Fig. 3b). Thus, Clones 2 and 6 Fab molecules are capable of recognizing SARS-CoV-2 S trimer in all possible RBD

**Table 1 Cryo-EM data collection and refinement statistics.**

| | 3dSpike-Fab6 (PDB 7MW2) (EMD-24060) | 2dSpike-Fab6 (PDB 7MW3) (EMD-24061) | 1dSpike-Fab6 (PDB 7MW4) (EMD-24062) | 2uSpike-Fab2 (PDB 7MW5) (EMD-24063) | 3uSpike-Fab2 (PDB 7MW6) (EMD-24064) |
|---|---|---|---|---|---|
| **Data collection and processing** | | | | | |
| Magnification | 81,000 | 81,000 | 81,000 | 81,000 | 81,000 |
| Voltage (kV) | 300 | 300 | 300 | 300 | 300 |
| Electron exposure (e–/Å$^2$) | 66.5 | 66.5 | 66.5 | 66.5 | 66.5 |
| Defocus range (μm) | −0.8 to −1.8 | −0.8 to −1.8 | −0.8 to −1.8 | −0.8 to −1.8 | −0.8 to −1.8 |
| Pixel size (Å) | 1.068 | 1.068 | 1.068 | 1.068 | 1.068 |
| Symmetry imposed | C3 | C1 | C1 | C1 | C1 |
| Initial particle images (no.) | 655318 | | | 1013040 | |
| Final particle images (no.) | 68416 | 111449 | 80954 | 135336 | 181380 |
| Map resolution (Å) FSC threshold | 2.97 | 3.15 | 3.42 | 3.42 | 3.22 |
| **Refinement** | | | | | |
| Map sharpening B factor (Å$^2$) | −81 to −91 | −69 to −127 | −76 to −142 | −74 to −189 | −80 to −100 |
| Model map FSC (masked) | 0.83 | 0.81 | 0.74 | 0.67 | 0.69 |
| Model composition | | | | | |
| Non-hydrogen atoms | 33,807 | 33,825 | 33,741 | 33,661 | 35,247 |
| Protein residues | 4293 | 4293 | 4290 | 4245 | 4446 |
| Ligands | 72 | 72 | 74 | 83 | 87 |
| B factors (Å$^2$) | | | | | |
| Protein | 41 | 46 | 63 | 118 | 39 |
| Ligand | 80 | 69 | 94 | 156 | 62 |
| R.m.s. deviations | | | | | |
| Bond lengths (Å) | 0.003 | 0.003 | 0.003 | 0.003 | 0.004 |
| Bond angles (°) | 0.6 | 0.6 | 0.6 | 0.7 | 0.8 |
| Validation | | | | | |
| MolProbity score | 1.6 | 1.7 | 1.8 | 2.2 | 2.0 |
| Clashscore | 7.1 | 7.5 | 8.7 | 14.1 | 11.0 |
| Poor rotamers (%) | 0.08 | 0.08 | 0.11 | 0.08 | 0.11 |
| Ramachandran plot | | | | | |
| Favored (%) | 96.68 | 96.09 | 95.64 | 91.39 | 94.27 |
| Allowed (%) | 3.15 | 3.55 | 4.15 | 8.03 | 5.06 |
| Disallowed (%) | 0.17 | 0.36 | 0.21 | 0.57 | 0.66 |

*3dSpike-Fab6* Spike trimer with three RBDs down in complex with Fab clone 6, *2dSpike-Fab6* Spike trimer with two RBDs down in complex with Fab clone 6, *1dSpike-Fab6* Spike trimer with one RBD down in complex with Fab clone 6, *2uSpike-Fab2* Spike trimer with two RBDs up in complex with Fab clone 2, *3uSpike-Fab2* Spike trimer with three RBDs up in complex with Fab clone 2.

conformations, potentially reducing viral immune evasion when used in combination or as a bispecific. Within each clone, the same Fab-RBD-binding interface is maintained regardless of the RBD conformations across all S trimer states. Between the two clones, the Fab-RBD interfaces are different, although the two Fab clones bind RBD in similar locations with RBD adopting virtually the same conformation (RMSD 0.46 Å) (Fig. 3a, b). Clone 2 Fab-S trimer complexes appear more flexible than Clone 6 complexes, as indicated by less well-defined cryo-EM density in the Clone 2 Fab-RBD regions.

Besides the major Fab-RBD interfaces described above, in the Clone 6 Fab complexes with two or three RBDs down, a down-RBD-binding Fab makes an additional contact to a side surface of an adjacent down-RBD to lock both RBDs in the closed conformation incompetent in ACE2 recognition (Supplementary Fig. 4d left two panels), which may provide a second neutralization mechanism. The additional contact is achieved by the CDRL1 loop of Clone 6 Fab that protrudes in between the two neighboring down-RBDs (Supplementary Fig. 4c), along with framework region 3 (FWR3) of the Fab that interacts with a nearby side surface of the adjacent down-RBD (Supplementary Fig. 4d, left two panels). In addition, in all complexes with two up/one down-RBDs and the Clone 6 Fab complex with two down/one up-RBDs, a down-RBD-binding Fab contacts the side surface of an adjacent up-RBD through another conserved

interface mainly involving the CDRL1 loop of the Fabs (Supplementary Fig. 4b, d, right panels), further stabilizing the conformation of the adjacent up-RBDs. These types of bivalent interactions are not observed for the up-RBD-binding Fabs.

Clone 2 and Clone 6 Fabs interact with spike RBD at similar locations, as both directly block access of the host receptor ACE2 (Fig. 3a). Clone 6 Fab shields the entire top surface of spike RBD (both "left shoulders" and "right ridge"), whereas Clone 2 Fab has a relative rotation of ~17° pivoted on the RBD right ridge, resulting in a larger contact area there and less contact with the RBD left shoulder (Fig. 3b). When overlaying a Clone 2 Fab-bound down-RBD onto a Clone 6 Fab-bound down-RBD in the S trimer with more than one RBDs down, spatial clashes are detected between the Clone 2 Fab and the neighboring down RBD (Supplementary Fig. 4a). This also explains why we did not observe Clone 2 Fab-bound S trimer in a conformation state with more than one RBD down. In each case, all six complementarity-determining regions (CDRs) of the Fab participate in RBD interactions (Fig. 3c). The three CDRH loops of the two clones share similar overall conformations and interact mainly with the right ridge of spike RBD (Fig. 3c, upper panel). The three CDRH loops of Clone 2 Fab form a large positively charged paratope, complementing a negatively charged epitope contributed by the right ridge of spike RBD (Supplementary Fig. 5a, upper panels). The three CDRH loops of Clone 6 Fab engage spike RBD

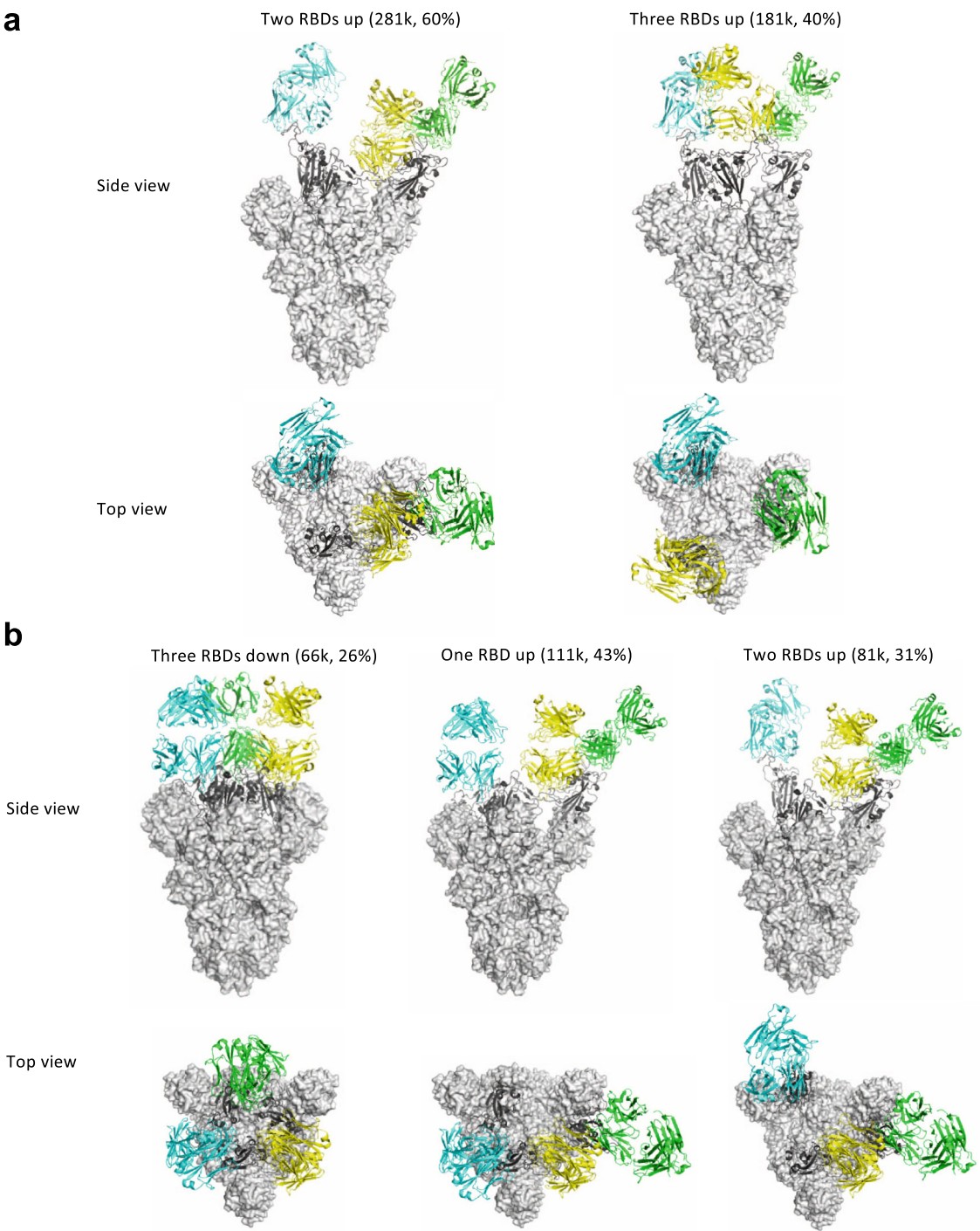

**Fig. 2 Cryo-EM structures of the ectodomain of SARS-CoV-2 spike trimer in complex with Clone 2 or Clone 6 Fab and Structural comparisons.**
**a**, **b** Cryo-EM structures of the ectodomain of SARS-CoV-2 spike trimer in complex with Clone 2 (**a**) or Clone 6 (**b**). Each Fab molecule is shown as ribbons in different colors, spike RBD is shown as dark gray ribbons, with the rest of spike trimer shown as gray surface. The particle distribution of each S trimer conformation is shown correspondingly.

predominantly through hydrophobic interactions (Supplementary Fig. 5b, upper left), with the CDRH3 loop also engaging RBD through modest electrostatic interactions (Supplementary Fig. 5b, upper right).

In contrast to the relatively conserved binding locations by the CDRH loops, the three CDRL loops of Clones 2 and 6 diverge in RBD-binding paratopes (Fig. 3c, lower panel). CDRL1 and CDRL3 of Clone 2 Fab primarily contact the epitopes at RBD right ridge and the CDRL2 loop engages RBD central groove,

whereas CDRL3, CDRL1, and CDRL2 of Clone 6 Fab complement with the RBD right ridge, central groove, and left shoulder, respectively, together covering the top surface of the RBD. Both Clones 2 and 6 Fab CDRL loops contact spike RBD primarily through hydrophobic interactions along with modest electrostatic interactions (Supplementary Fig. 5a, b, lower panels). A comparison of the Clone 2 and Clone 6 Fabs revealed that the most different region comes from the CDRH3 loop (Fig. 3d). The Clone 6 Fab contains a longer CDRH3 loop, which inserts into

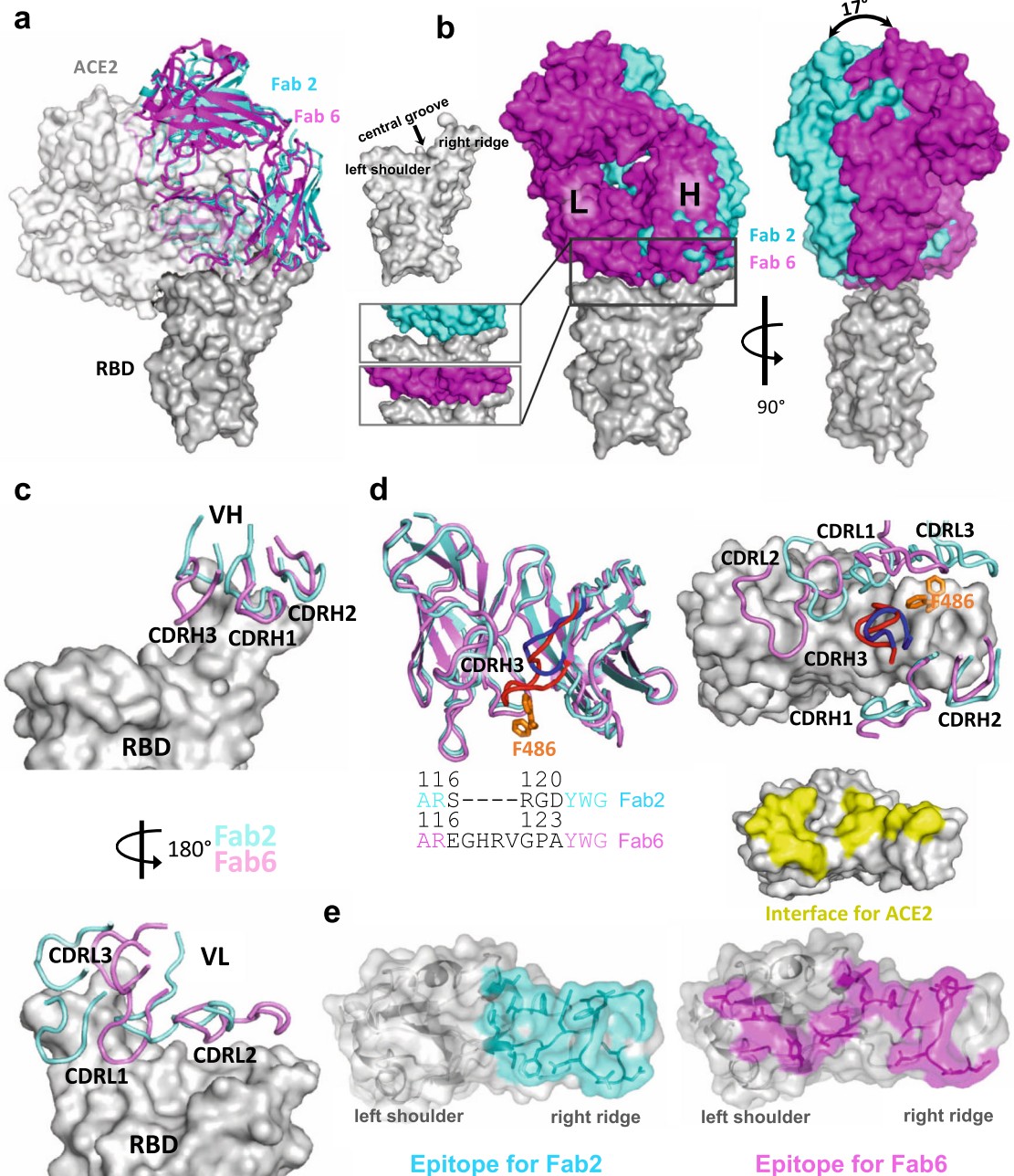

**Fig. 3 Binding interfaces and epitopes of lead monoclonal antibodies. a** Overlay of the structures of spike RBD from its complex with the host ACE2 receptor, Clone 2 or Clone 6 Fab indicates both antibody clones directly block the ACE2 access. **b** Overlay of the structures of spike RBD from its complex with Clone 2 or Clone 6 Fab reveals a 17° rotation pivoted at the right ridge of spike RBD, resulting in less contact of Clone 2 Fab light chain with the left shoulder of spike RBD. The comparison of the Clone 2 and Clone 6 Fab binding interfaces with spike RBD is shown in the bottom parallel insets. The nomenclature of the top surface of spike RBD is described in the top inset. **c** The orientations of the three CDRH (upper panel) and CDRL (lower panel) loops of Clone 2 or Clone 6 Fabs on spike RBD. The structures were overlaid with the RBD potions. **d** Left, overlay of Clone 2 and Clone 6 Fab molecules alone. The CDRH3 loops that have the largest conformational difference are highlighted in brighter colors (blue for clone 2 and red for clone 6). Lower inset, amino acid sequence alignment of the CDRH3 loops of both clones. Right, spike RBD residue F486, which is shown as orange sticks, has a large conformational change upon RBD-binding to the two clones. **e** Comparison of spike RBD epitope regions for Clone 2 and Clone 6 Fabs, highlighted in cyan and magenta, respectively. The spike binding interface for the hACE2 receptor is highlighted in yellow.

the central groove of spike RBD to generate a tighter engagement. As a consequence, this induces a substantial change in the RBD, which otherwise maintained a nearly identical conformation, where residue F486 flips to contact the CDRL3 loop of Clone 6 instead of interacting with the CDRH3 loop of Clone 2 (Fig. 3d). Even though the two Fabs interact with the RBD with similar buried surface areas (949 Å² in Clone 2 vs. 894 Å² in Clone 6), the

further spread-out RBD-binding mode of Clone 6 Fab is more similar to that of the host ACE2 receptor (Fig. 3e).

Multiple mAbs have been generated to date and are in various stages of research and development, with some granted EUA for clinical use[3,28,43–45]. We compared our structures to several of the previously published mAb:Spike structures, and found that the RBD-binding modes of Clone 2 and Clone 6 Fabs differ from

these SARS-CoV-2 neutralizing antibodies reported in the primary literature and the protein databank (PDB). Three previously reported antibodies (2H2, CV05-163, and S2H13) target the spike RBD in somewhat similar orientations but with substantial rotations or shifts[46–48] (Supplementary Fig. 6a). Another two reported antibodies (CT-P59 and BD23) adopt RBD-binding conformations resembling those of Clone 2 and 6 Fabs[27,49], however, the binding positions of heavy chains and light chains are exchanged (Supplementary Fig. 6b).

**Lead antibody clones potently neutralize SARS-CoV-2 WA1 and B.1.617 variant.** We next tested the neutralization ability of the monospecific and bispecific clones. Because spike protein can mediate receptor-mediated membrane fusion[50], we utilized a cell fusion assay in which expression of Wuhan-1/WA1 SARS-CoV-2 spike on one cell can induce fusion with hACE2-expressing cells. All three antibodies (Clone 2, Clone 6, and Clone 16) effectively inhibited this spike-mediated cell fusion activity (Supplementary Fig. 2d, e). Next, we used an HIV-1-based pseudovirus system pseudotyped with SARS-CoV-2 Wuhan-1/WA1, B.1.351, and B.1.617 to evaluate the neutralization potential of these mAbs. All three antibodies (Clone 2, Clone 6, and Clone 16) potently inhibited SARS-CoV-2-Wuhan-1 pseudovirus with low ng/mL level $IC_{50}$ values (Fig. 4a). Due to the partial overlap in the binding domains in RBD, the bispecific Clone 16 was not stronger than either Clone 2 or Clone 6. These three antibodies also inhibited B.1.351 SARS-CoV-2 pseudovirus, with somewhat reduced potency (higher level of $IC_{50}$ values, mid ng/mL for Clones 2/6, and high ng/mL for Clone 16) (Fig. 4b). In comparison, all three antibodies maintained potent neutralizing activity against the B.1.617 pseudovirus, with sub-single-digit to low-single-digit ng/mL level $IC_{50}$ values (Fig. 4c). We also performed these experiments with Clones 2 and 6 as a cocktail combination (Fig. 4d–f). Although the combination is not superior to the single clone alone, the results again confirmed the potency of both Clones 2 and 6 as the single agents, where both clones showed strong potency against Wuhan-1/WA1 spike and B.1.617 pseudoviruses (single-digit ng/mL $IC_{50}$) and slightly reduced potency against B.1.351 (Fig. 4d–f).

**Epitope and mutation analysis of the mAb:RBD structures on the binding interface between the lead mAbs and hotspot mutations in Beta and Delta variants.** To evaluate the structural impacts of the RBD mutations on the binding of the Beta and Delta variants, we used our Spike:mAb Cryo-EM structures and generated homology models of the two variants (Fig. 4g). The Beta variant encodes three mutations (K417N, E484K, and N501Y) in spike RBD. Although N501 is not located within the epitope region, the Y501 mutation may have weak interactions with the CDRL2 loop of both Fab clones (Fig. 4g, lower left). K417 has no contact with Clone 2 Fab, but interacts with the CDRL1 loop of Clone 6 Fab, and the K417N mutation would likely abolish this interaction (Fig. 4g, lower left). E484, which is located on top of the RBD right ridge, is targeted by the CDRH1 and CDRH3 loops of both Fab clones and has a larger contact area on Clone 6 Fab (Fig. 4g, upper left). The Delta variant encodes an E484Q at the position, which likely results in less disruption of its interaction with the Fabs (Fig. 4g, upper right). The two other mutation sites, L452 and T478, are not located within the antibody-binding surface (Fig. 4g, right). However, the T478K mutation is located close to the interface, especially that with Clone 2 Fab, and could affect RBD recognition by both clones (Fig. 4g, lower right).

**In vivo prophylactic and therapeutic efficacy of the lead mAbs against authentic SARS-CoV-2 virus.** We next evaluated the potency of the lead antibodies against authentic SARS-CoV-2 (WA1/2020) infection. All three antibodies (Clone 2, Clone 6, and the bispecific Clone 16) inhibited infection of SARS-CoV-2 WA1/2020 (low-mid ng/mL level IC50s) (Fig. 5a). We then assessed the efficacy of mAbs against SARS-CoV-2 in vivo, when administered as either pre-exposure prophylaxis or postexposure therapy (Fig. 5b). We performed protection studies with SARS-CoV-2 using K18-hACE2 transgenic mice[51–53]. We challenged K18-hACE2 mice with $2 \times 10^3$ plaque-forming units (PFU) of the SARS-CoV-2 WA1/2020 virus (Fig. 5b). The hACE2 transgenic mice were randomly divided into three groups, and mice in each group were injected with 20 mg/kg of Clone 2 mAb, Clone 6 mAb, or placebo control; with the treatment given as a single dose either 24 h before or 18 h after viral infection (Fig. 5b).

In the prophylactic setting, all (8/8, 100%) mice in the placebo group developed a severe disease due to viral challenge, and most (7/8, 87.5%) of them lost substantial body weight and succumbed from the disease, with only one mouse recovering weight (Fig. 5c–e). In contrast, all mice receiving treatment of either Clone 2 (8/8, 100%) or Clone 6 (7/7, 100%) maintained their body weight throughout the duration of the study and survived SARS-CoV-2 infection (Fig. 5c–e). In the therapeutic setting, all (5/5, 100%) mice in the placebo group developed the severe disease, and most (4/5, 80%) of them lost body weight (Fig. 5f–h). In contrast, all mice treated with Clone 2 (5/5, 100%) or Clone 6 (5/5, 100%) at +18 h maintained their body weight and survived the viral infection with no signs of disease (Fig. 5f–h). These data suggested that these mAbs can protect the animals from lethal SARS-CoV-2 infection in either prophylactic or therapeutic settings.

**Generation, biophysical characterization, and functional testing of a humanized mAb clone.** To improve clinical translatability, we also developed a humanized Clone 2, using standard antibody humanization approaches with framework humanization and engineered mutations, based on canonical human antibody backbone sequences as well as the antibody: RBD Cryo-EM structures (Methods). A resultant humanized clone that maintains RBD specificity was generated (Clone 13 A) (Fig. 6a). We purified Clone 13 A with other lead clones for characterization and functional studies (Fig. 6b). BLI data showed that Clone 13 A has a low-double-digit Kd value (Fig. 6c).

Clone 13 A also potently neutralized WT/WA1 and B.1.617 pseudoviruses (single-digit ng/mL $IC_{50}$) (Fig. 6d). We also performed authentic virus neutralization assays with Clone 13 A, along with Clone 2, Clone 6, and two EUA antibodies (RGEN 10933 and 10987). Clones 2, 6, and 13 A all potently neutralized SARS-CoV-2 WA1/2020 and Delta variant authentic viruses (Fig. 6e). Against Delta variant, RGEN 10933 has a 1.6× drop in potency, Clone 13 A has a 2× drop, Clone 2 has a 4× drop, Clone 6 has an 18× drop, while RGEN-10987 has a 52× drop.

We also performed in vivo challenge experiment with Clone 13 A using authentic viruses (Fig. 6f–h). Similar to the prior results all (8/8, 100%) mice in the placebo group developed a severe disease due and succumbed from infection (Fig. 6f, h). In contrast, all mice receiving Clone 13 A (9/9, 100%) maintained body weight throughout the duration of the study and survived from SARS-CoV-2 WA1 infection (Fig. 6g, h). We also tested Clone 13 A against the Delta variant authentic virus challenge. All (8/8, 100%) mice in the placebo group developed a severe disease due and succumbed from infection (Fig. 6i). In contrast, all mice receiving Clone 13 A (8/8, 100%) maintained body weight throughout the duration of the study (Fig. 6j). Of note, the Delta

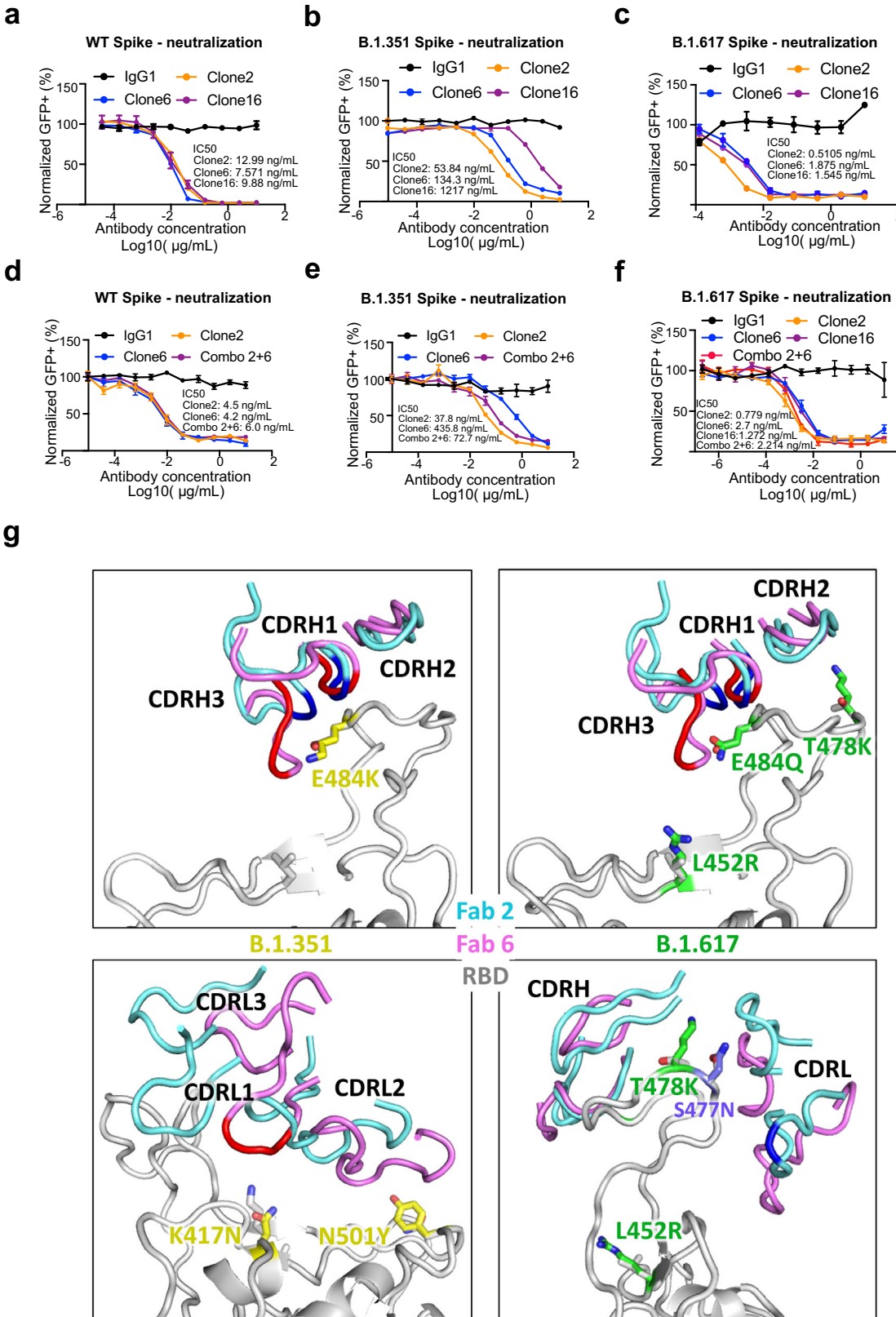

virus appeared to be not lethal for hACE2 mice within the experimental setting, thus the survival benefit can not be measured against Delta. These data confirmed that the humanized Clone 13 A maintained its potency and protective activity in vivo against authentic SARS-CoV-2, both the original WA1 and Delta variant in the B.1.617 lineage.

## Discussion

The ongoing COVID-19 pandemic and the continued emergence of SARS-CoV-2 variants have necessitated the rapid development of therapeutic interventions. The discovery and development of neutralizing antibodies with expanding collections of epitopes are critical to provide countermeasure options against escape seen

**Fig. 4 Lead mAb clones showed strong neutralization potency against WT/WA1 and B.1.617, with epitope and mutation analysis of the Fab-spike RBD interfaces. a** Neutralization assay on top mAb clones using WT/WA1 SARS-CoV-2 Spike pseudotyped HIV-1-lentivirus carrying an EGFP reporter (pseudovirus). Data were presented as mean values ± SEM, $n = 3$ biological replicates. **b** Neutralization assay on top mAb clones using B.1.351 variant SARS-CoV-2 Spike pseudovirus. Data were presented as mean values ± SEM, $n = 3$ biological replicates. **c** Neutralization assay on top mAb clones using B.1.617 variant SARS-CoV-2 Spike pseudovirus. Data were presented as mean values ± SEM, $n = 3$ biological replicates. **d** Neutralization assay on top mAb clones and their combination using WT/WA1 SARS-CoV-2 Spike pseudovirus. Data were presented as mean values ± SEM, $n = 3$ biological replicates. **e** Neutralization assay on top mAb clones and their combination using B.1.351 variant SARS-CoV-2 Spike pseudovirus. Data were presented as mean values ± SEM, $n = 3$ biological replicates. **f** Neutralization assay on top mAb clones and their combination using B.1.617 variant SARS-CoV-2 Spike pseudovirus. Data were presented as mean values ± SEM, $n = 3$ biological replicates. **g** Structural analysis of the spike RBD mutations from the B.1.351 (left panels) and B.1.617 (right panels) variants at the interfaces with Clone 2 and Clone 6 Fabs. The residues mutated in spike RBD are shown as gray sticks in wild-type (WT) spike and yellow in the B.1.351 variant and green in the B.1.617 variant. Another frequent RBD mutation S477N in SARS-CoV-2 strains is also shown as slate sticks. The spike RBD is shown as a light gray ribbon, and the CDR loops of Clone 2 and Clone 6 Fabs are shown as cyan and magenta ribbons, respectively, with the WT RBD-interacting residues, highlighted in brighter blue and red colors. Source data and additional statistics for experiments are provided as a Source Data file.

with emerging variants of concern. We combined SARS-CoV-2 Spike RBD protein immunization with high-throughput single-cell BCR sequencing technology to establish a platform to develop neutralizing antibody candidates. We identified two highly potent and specific SARS-CoV-2 neutralizing mAb clones with single-digit nanomolar affinity and low-picomolar avidity. We also generated a bispecific antibody of these two lead clones, as well as a potent humanized clone. The lead antibodies showed strong neutralization ability against SARS-CoV-2 and the highly transmissible B.1.617 lineage that poses a risk of reducing the efficacy of currently available therapeutic antibodies and prophylactic vaccines.

The SARS-CoV-2 spike protein is dynamic, and three conformations of spike pre-fusion trimers have been detected on intact virions: all RBDs down, 1 RBD up, and 2 RBDs up, with the last one only existing in vitro with multiple stabilizing mutations[54]. The spike RBD needs to extend upwards to be accessible to ACE2, and the presence of host ACE2 changes the population distribution of different spike conformations by promoting the RBD towards the up/open state favorable for ACE2 binding[55]. Genetic variations in spike can also change the conformational equilibrium of the trimer, subsequently affecting virus infectivity[56]. Although antibody therapy has been developing rapidly to prevent or treat SARS-CoV-2, infection, the precise determinants of neutralization potency remain unclear. In addition to direct receptor blockade by antibody binding, modulation of spike-mediated membrane fusion by altering the ACE2-triggered spike protein conformational cycle has been suggested as another determinant of the antibody neutralization potency[50].

Depending on the binding mode, some antibodies may facilitate the spike conformational cycle to the final stage with all three RBDs open, or locking the spike trimer in a pre-fusion state, therefore enhancing or inhibiting the cell membrane fusion and syncytium formation. Based on our cryo-EM analysis, Clone 6 Fab binds to spike trimer preferentially with at least one RBD down and effectively skews the spike trimer towards pre-fusion states (Fig. 2b). Besides making two neighboring down-RBDs inaccessible for ACE2 binding (Supplementary Fig. 4c, d, left panels), a down-RBD-binding Clone 6 Fab can also interact with an adjacent up-RBD through a quaternary epitope located at the sidewall of the up-RBD (Supplementary Fig. 4d, right), which clashes with ACE2 binding (Supplementary Fig. 4e, right), thereby directly blocking ACE2 access to two RBDs simultaneously. This bipartite binding mode presumably is more stable than the single binding mode with the up-RBD alone, explaining why no spike trimer with all three RBDs open was detected when complexed with Clone 6 Fab. We hypothesize that the Clone 6 Fab binding to this secondary epitope of RBD helps lock the spike

trimer in the pre-fusion form, which inhibits the spike-mediated cell membrane fusion by historical virus and even the B.1.1.7 variant that has enhanced binding affinity to ACE2[57] (Supplementary Fig. 2d, e, middle). In contrast, due to the different binding conformations of Clone 2 Fab on spike RBD, the spike trimer has been detected in skewed states that favorite RBDs up (Fig. 2a), which mimics the effect of ACE2 binding during the conformational cycle of the spike trimer. Nonetheless, we found that Clone 2 Fab also efficiently suppresses spike-mediated cell-cell fusion[50] (Supplementary Fig. 2d, e, left), suggesting that the mechanism determining the neutralization potency of spike-targeting antibodies is complex.

Antibody resistance of SARS-CoV-2 B.1.1.7, B.1.351, and B.1.617 lineage variants has been reported[7,14,58]. The B.1.1.7 variant of the SARS-CoV-2 spike contains a single mutation (N501Y) in the RBD, which has been reported to enhance the spike RBD-ACE2 binding affinity that could disfavor antibody neutralizations competing with ACE2 for RBD-binding[57]. N501 does not directly interact with either Clone 2 or Clone 6 Fab, although the mutation may generate allosteric effects on other CDRL loops nearby to disrupt the binding interface[57]. While the B.1.1.7 variant spreads faster, has a higher case-fatality rate, and has some antibody resistance, it does not reduce the efficacy of the currently approved vaccines[7,59–61]. In contrast, the B.1.1.351 variant of SARS-CoV-2 showed an increased rate of transmission, resistance to antibody therapeutics, and reduced vaccine efficacy[6–8,60]. In addition to the N501Y mutation, the B.1.351 variant has two additional point mutations K417N and E484K in spike RBD, which perturbs the RBD epitope recognition by both antibody clones, explaining their reduced potencies against the B.1.351 variant. The lineage B.1.617 has mutations in spike including G142D, E154K, L452R, E484Q, D614G, P681R, and Q1071H[15], which could affect a number of leading therapeutic antibodies tested to date[14,62,63]. The L452R variant evades cellular immunity and increases infectivity[64]. L452R and S477N might affect the potency of Clones 2 and 6 to some degree based on the structure.

Several therapeutic antibodies have been granted EUA for clinical use by the FDA, such as two from Regeneron developed in a previous study[65]. Recent studies showed that some of the EUA mAbs have a significant reduction in neutralization activities against B.1.617 lineage variants[14,62]. Many antibodies have been developed by the field and tested against various VoCs[63,66,67]. Our data also showed that RGEN-10987 has substantially diminished potency against Delta. To better understand the impact of Delta mutations on antibody neutralization potency, we performed structurally; analysis with the Delta RBD (Supplementary Fig. 8). The epitopes of REGN-10987 centered around the RBD left shoulder loop on the β1 end, which hereafter

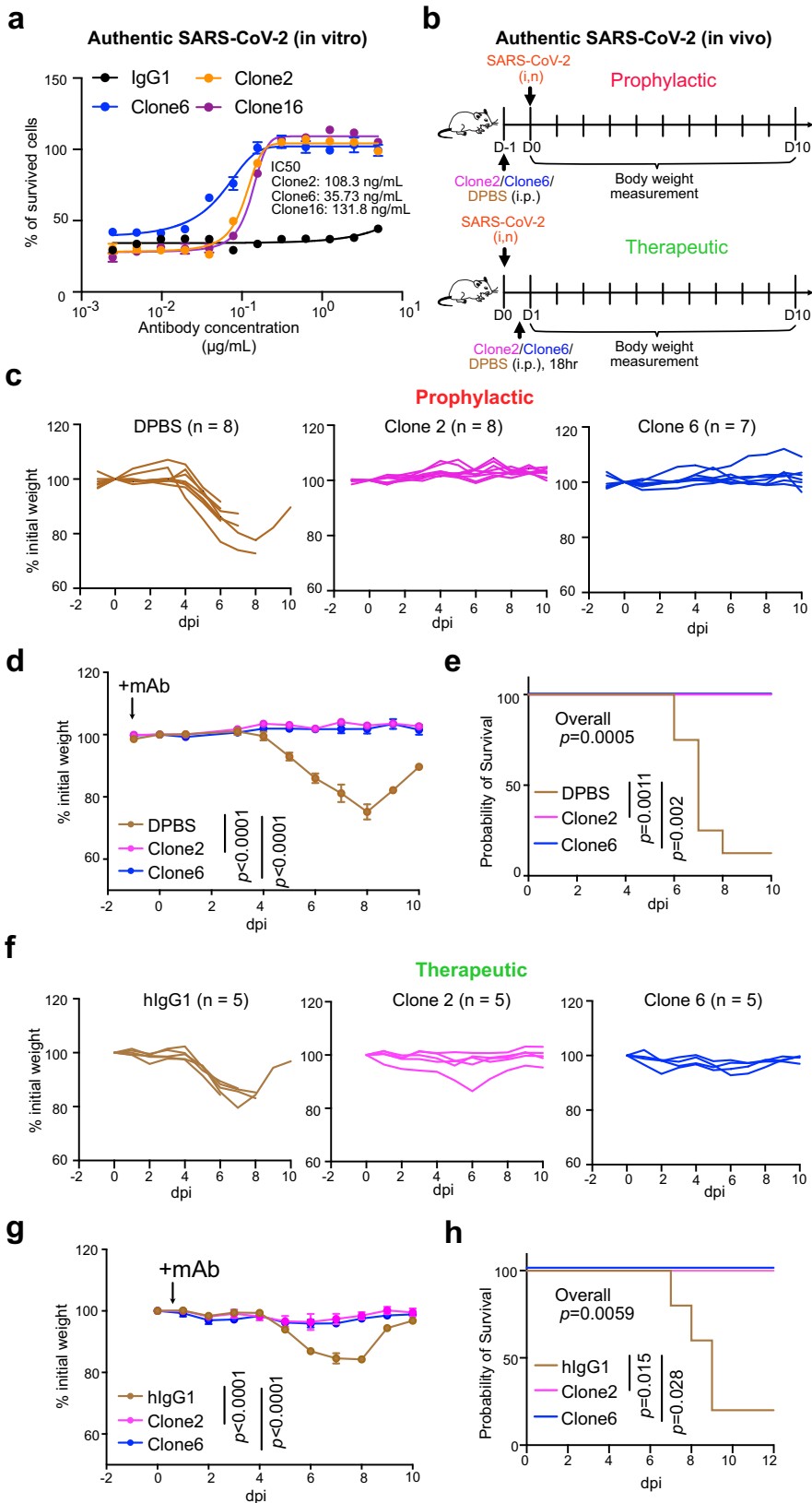

we termed left shoulder loop 1 (Supplementary Fig. 8). Clone 2 primarily targets a region at the right ridge of RBD. The L452R mutation of Delta is on the C-terminus edge of the left shoulder loop 1 and is critical for the loop 1 conformation. L452R is observed to favor an orientation interacting with the Y351 and is likely to disrupt the original conformation of left should loop 1,

consistent with the reduction of the affinity and potency of the REGN-10987 antibody. The E484Q, on the other hand, was not observed to significantly change side-chain orientation and is less likely to cause drastic conformational changes on nearby loops. The mutation-induced side-chain orientation, as well as nearby loop conformation change plus distinct antibody targeting

**Fig. 5 Lead monoclonal antibodies have potent prophylactic and therapeutic in vivo efficacy against replication-competent SARS-CoV-2 virus. a** In vitro neutralization of top mAb clones against authentic SARS-CoV-2 in BL3 setting, $n = 3$ biological replicates. **b** Schematics of in vivo efficacy testing of top mAb clones against lethal challenges of authentic SARS-CoV-2 in hACE2 transgenic mice at BL3 level, in both prophylactic (upper panel) and therapeutic (bottom panel) settings. **c–e** Prophylactic efficacy testing. **c, d** Body weight curves of antibody and placebo (DPBS) treated hACE2 transgenic mice under lethal challenges of authentic SARS-CoV-2. **c** Spider curves of body weight changes of individual mouse plotted separated by group. **d** Mean ± SEM curves of body weight changes. (Clone 2, $n = 8$; Clone 6, $n = 7$, and DPBS, $n = 8$ mice, respectively), Clone 2 vs DPBS: $p < 0.0001$, Clone 6 vs DPBS: $p < 0.0001$. **e** Survival curves of antibody and placebo (DPBS) treated hACE2 transgenic mice under lethal challenges of authentic SARS-CoV-2 (the same experiment in **c, d**), a two-sided statistical test was used to assess statistical significance. Therapeutic efficacy testing. **f–h** Body weight curves of antibody and placebo (isotype hIgG1) treated hACE2 transgenic mice under lethal challenges of authentic SARS-CoV-2. **f** Spider curves of body weight changes of individual mouse plotted separated by group. **g** Mean ± SEM curves of body weight changes. (Clone 2, $n = 5$; Clone 6, $n = 5$, and hIgG1, $n = 5$ mice, respectively), Clone 2 vs hIgG1: $p < 0.0001$, Clone 6 vs hIgG1: $p < 0.0001$. **h** Survival curves of antibody and placebo (hIgG1) treated hACE2 transgenic mice under lethal challenges of authentic SARS-CoV-2 (the same experiment in **f, g**), a two-sided statistical test was used to assess statistical significance. In this figure: Data were shown as mean ± s.e.m. plus individual data points in dot plots. Statistics: Two-way ANOVA was used to assess statistical significance for multi-group curve comparisons; Log-rank test was used to assess statistical significance for survival curve comparisons; unless otherwise noted. The $p$ values are indicated in the plots. Source data and additional statistics for experiments are as a Source Data file.

---

epitopes, are consistent with the observation that Clone 2 retained its neutralization potency against Delta, whereas REGN-10987 did not.

Here, we show, our lead antibody clones are distinctly different from existing antibodies reported to date in their binding geometry and footprints[68], are highly inhibitory against SARS-CoV-2 and several variants of concerns, particularly the Delta variant from the B.1.617 lineage. These antibodies thus expand the repertoire of COVID-19 countermeasures against the SARS-CoV-2 pathogen and its emerging and potentially more dangerous variants.

## Methods

**Institutional approval**. This study has received institutional regulatory approval. All recombinant DNA (rDNA) and biosafety work were performed under the guidelines of the Yale Environment, Health and Safety (EHS) Committee with approved protocols (Chen-15-45, 18-45, 20-18, 20-26; Xiong-17302; Wilen-18/16-2). All animal work was performed under the guidelines of Yale University Institutional Animal Care and Use Committee (IACUC) with approved protocols (Chen-2018-20068; Chen-2020-20358; Wilen-2018-20198).

**Animal immunization**. Standard 28-day repetitive immunization protocol was utilized for immunization. *M. musculus* (mice), 6–12 weeks old females, of C57BL/6 J and BALB/c strains, were purchased from Jackson laboratory and used for immunization. First, all mice were ear-marked and around 200 μl blood was taken as a pre-immunization sample, where serum was collected from the blood by centrifugation (1000×*g* for 10 min). Two days later (day 0), for each mouse, 20 μg SARS-CoV-2 RBD-his tag protein (Sino biological) in 100 μl PBS was mixed with 100 μl Complete Freund's Adjuvant (CFA) with three-way stop Cock. The fully emulsified mixture was subcutaneously injected into the back of each mouse. On day 7, a second immunization was performed, where each mouse was injected subcutaneously with 20 μg RBD-his tag protein fully emulsified with Incomplete Freund's Adjuvant (IFA). On day 13, around 50 μl of blood from each mouse was obtained for serum preparation as first bleeds. On day 14, a third immunization is performed, where all the procedures were similar to the second immunization. On day 20, second bleeds were taken. On day 21, the fourth immunization is performed, where all the procedures were similar to the second immunization. On day 24, each mouse receives 20 μg RBD-his tag protein in 200 μl PBS intraperitoneally as final immunization. On day 28, mice with strong serum conversion detected by ELISA were sacrificed. Spleen, lymph nodes, and bone marrow were collected for B cells isolation and purification for single-cell BCR sequencing. Serums from pre-, first, and second bleeds were subjected to ELISA for anti-RBD titer determination.

**Mouse B cell isolation and purification**. Primary B cells from the spleen, draining lymph nodes, bone marrow of RBD-his tag protein immunized mice were isolated and purified with mouse CD138 MicroBeads (Miltenyi Biotec, 130-098-257) following the standard protocol provided by the manufacturer. Spleens and draining lymph nodes were homogenized gently. Bone marrows were fragmented, rinsed with PBS containing 2% FBS, and filtered with a 100 μm cell strainer (BD Falcon, Heidelberg, Germany). The cell suspension was centrifuged for 5 min with 400 × *g* at 4 °C. Erythrocytes were lysed briefly using ACK lysis buffer (Lonza) with 1 mL per spleen for 1–2 min before adding 10 mL PBS containing 2% FBS to restore iso-osmolarity. The single-cell suspensions were filtered through a 40 μm cell strainer (BD Falcon, Heidelberg, Germany). CD138 positive B cells were isolated using magnetic cell sorting by positive selection according to the manufacturer's

instructions. Cell samples post-magnetic selection were counted and prepared for single-cell BCR sequencing.

**Single-cell BCR sequencing**. The enriched CD138+ plasma cells and progenitor B cells were loaded on a 10X Chromium Next GEM Chip G. The target cell number was 10,000 cells per sample. Single-cell lysis and RNA first-strand synthesis were performed using Chromium Next GEM Single Cell 5′ Gel Bead V2 according to the manufacturer's protocol. The following RNA and V(D)J library preparation was performed according to the manufacturer's protocol (Chromium Next GEM Single Cell V(D)J reagent kit, mouse BCR). The resulting VDJ-enriched libraries were sequenced following the reading mode recommended by 10× Genomics. Sequencing was performed on a NovaSeq targeted for 10,000 reads/cell, with a total of 100 million reads.

**Single-cell VDJ sequencing data analysis**. Raw sequencing data were processed using Cell Ranger v3.1.0 with default settings, aligning the reads to the GRCm38 mouse VDJ reference. Outputs from Cell Ranger were then visualized using the Loupe V(D)J Browser for quality control assessment and to identify the top enriched clonotypes. The consensus amino acid sequences for the top-ranked heavy/light chain pairs in each sample were then extracted and codon-optimized for human expression.

**Plasmid construction**. The cDNA sequences of the paired variable heavy and light chain region of anti-RBD antibody clones were synthesized as gBlocks (IDT) and cloned by the Gibson assembly (NEB) into human IgG1 heavy chain and light chain expression plasmids, pFUSEss-CHIg-hG1(InvivoGen, pfusess-hchg1) and pFUSE2ss-CLIg-hK (InvivoGen, pfuse2ss-hclk), respectively. pFUSEss-CHIg-hG1 plasmid is a cloning plasmid that expresses the constant region of the human IgG1 heavy chain and includes multiple cloning sites to enable cloning of the heavy chain (CH) variable region. Parallelly, pFUSE2-CLIg-hK is a cloning plasmid that expresses the constant region of the human kappa light chain and contains multiple cloning sites to enable cloning of the light chain variable region. For anti-RBD antibody clones' heavy chain plasmid cloning, gBlocks, containing cDNA sequence of the variable region of the heavy chain of anti-RBD antibody clones and the regions overlapping with corresponding flanking sequences of EcoRI and NheI restriction sites pFUSEss-CHIg-hG1, were ordered from IDT. pFUSEss-CHIg-hG1 were digested with EcoRI and NheI restriction enzyme (Thermo Fisher). These synthesized gBlocks were cloned into gel-purified restriction enzyme digested backbone by the Gibson assembly (NEB). For anti-RBD antibody clones' light chain plasmid cloning, gBlocks, containing cDNA sequence of the variable region of the light chain of anti-RBD antibody clones and the regions overlapping with corresponding flanking sequences of EcoRI and BsiWI restriction sites pFUSE2ss-CLIg-hK, were ordered from IDT. The gBlocks were then cloned into the pFUSE2ss-CLIg-hK backbone, which was digested with EcoRI and BsiWI restriction enzyme (Thermo Fisher).

The bispecific antibody with the same Fab regions of clone 2 and clone 6 was generated by using the CrossMab-KiH bispecific constructs[69]. The CrossMab-KiH bispecific constructs were designed and generated based on pFUSEss-CHIg-hG1 and pFUSE2ss-CLIg-hK. The bispecific antibody consists of two hetero-half IgG1, one is knob IgG1, and the other is hole IgG1 (Knob-in-Hole conformation). Four plasmids were employed: pFUSE2ss-knobLight-hK, pFUSE2ss-knobheavy-hG1, pFUSE2ss-HoleLight-hK, and pFUSE2ss-HoleHeavy-hG1. The pFUSE2ss-knobLight-hK is pFUSE2ss-CLIg-hK with no further editing. The pFUSE2ss-knobheavy-hG1 contains two knob mutations (T366W and S354C) in the CH3 region when compared with pFUSEss-CHIg-hG1. The gBlock (pPR024), containing constant region of heavy chain with two knob mutations and the regions overlapping with corresponding flanking sequences of NsiI and NheI restriction sites in pFUSEss-CHIg-hG1 was ordered from IDT and then cloned into

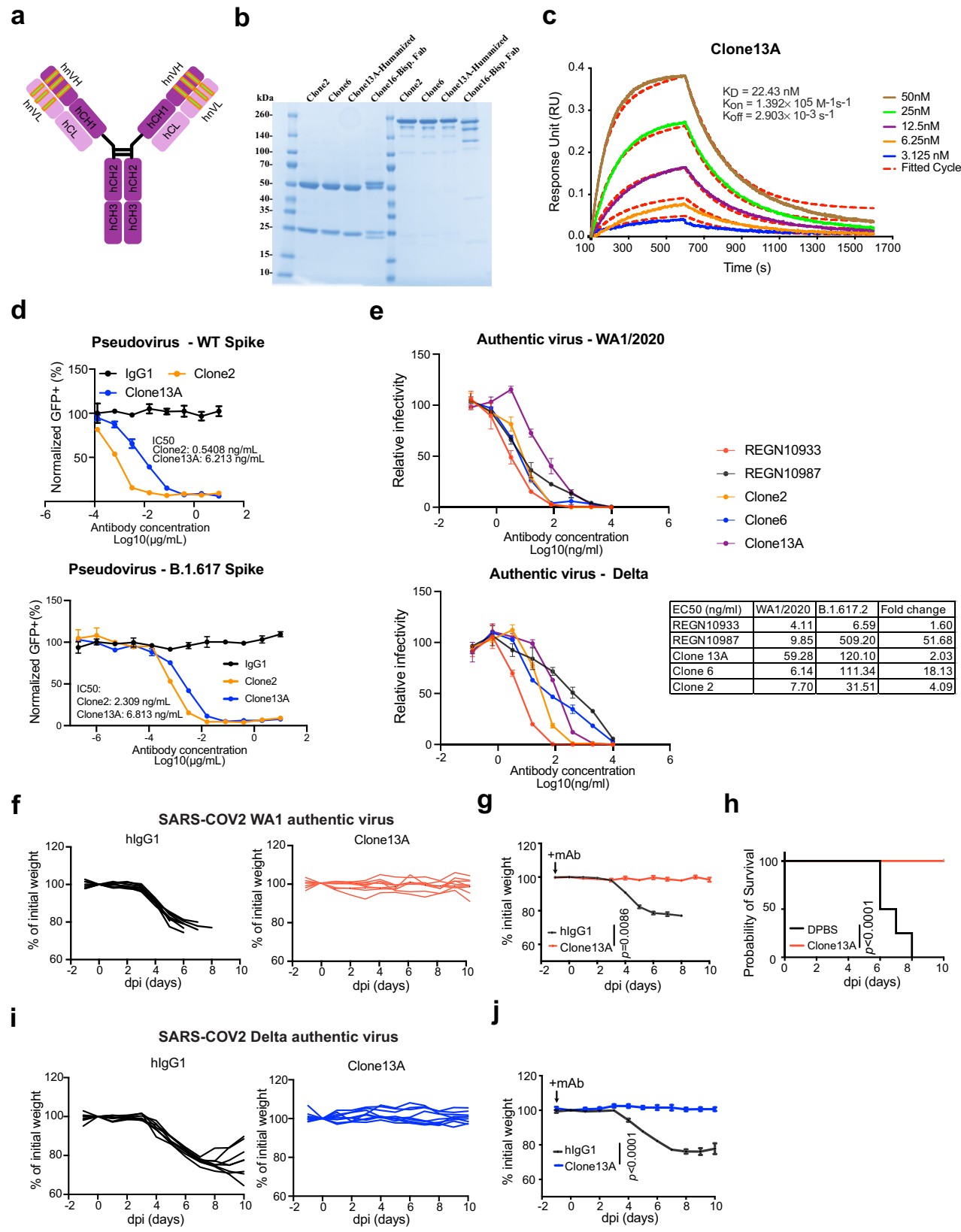

NsiI and NheI restriction enzymes digested pFUSEss-CHIg-hG1 backbone by the Gibson assembly (NEB). The pFUSE2ss-HoleLight-hK was generated by replacing the constant region of light chain (CL) in pFUSE2ss-CLIg-hK with CH1 region of heavy chain in pFUSEss-CHIg-hG1 vector. The CH1 region were PCR amplified from pFUSEss-CHIg-hG1vectors with a forward primer (oPR81-F) and a reverse primer (oPR82-R) containing regions overlapping with corresponding flanking sequences of the NcoI and NheI restriction sites in the pFUSE2ss-CLIg-hK. CH1

PCR amplified fragments were gel-purified and cloned into restriction enzyme digested pFUSE2ss-CLIg-hK by the Gibson assembly (NEB). The pFUSE2ss-HoleHeavy-hG1 possesses three "hole" mutations (T366S, L368A, and Y407V) in the CH3 region and a Y349C on the "hole" side to form a stabilizing disulfide bridge. In addition, to get the correct association of the light chain and the cognate heavy chain, the CH1 region in the pFUSE2ss-HoleHeavy-hG1 was exchanged with the constant region of the light chain (CrossMab conformation). The gBlock

**Fig. 6 Generation, biophysical characterization, and functional testing of a humanized mAb clone. a** Schematics of antibody humanization. **b** SDS-PAGE of purified antibodies of humanized clones (Clone 13A) along with other clones, this experiment had been done for once. **c** Octet measurement of Clone 13A using the BLI assay. **d** (Top) Neutralization assay on Clone 2 ($n = 3$ biological replicates) and a Clone 13A (humanized Clone 2) ($n = 3$ biological replicates) using WT SARS-CoV-2 Spike pseudotyped HIV-lentivirus carrying an EGFP reporter. (Bottom) Neutralization assay on Clone 2 ($n = 2$ biological replicates) and a Clone 13A (humanized Clone 2) ($n = 3$ biological replicates) using B.1.617 variant SARS-CoV-2 Spike pseudotyped HIV-lentivirus carrying an EGFP reporter. **e** In vitro neutralization of top mAb clones (Clones 2, 6, 13A) along with representative therapeutic antibody clones (RGEN 10933, 10987) against authentic SARS-CoV-2 WA1/2020 and B.1.617.2 (Delta) viruses in BL3 setting ($n = 4$ biological replicates). **f**, **g** Body weight curves of humanized antibody Clone 13 A and placebo (isotype hIgG1) treated hACE2 transgenic mice under lethal challenges of authentic SARS-CoV-2 WA1. **f** Spider curves of body weight changes of individual mouse plotted separated by group. **g** Mean ± SEM curves of body weight changes. (Clone 13 A, $n = 9$, and hIgG1, $n = 8$ mice, respectively), Clone 13 A vs hIgG1: $p = 0.0086$. **h** Survival curves of Clone 13 A antibody and placebo (hIgG1) treated hACE2 transgenic mice under lethal challenges of authentic SARS-CoV-2 (the same experiment in **f**, **g**), Clone 13 A vs hIgG1: $p < 0.0001$, a two-sided statistical test was used to assess statistical significance. **i**, **j** Body weight curves of humanized antibody Clone 13 A and placebo (isotype hIgG1) treated hACE2 transgenic mice under lethal challenges of authentic SARS-CoV-2 Delta. **i** Spider curves of body weight changes of individual mouse plotted separated by group. **j** Mean ± SEM curves of body weight changes. (Clone 13 A, $n = 8$, and hIgG1, $n = 8$ mice, respectively), Clone 13 A vs hIgG1: $p < 0.0001$. In this figure: Data were shown as mean ± s.e.m. plus individual data points in dot plots. Statistics: Two-way ANOVA was used to assess statistical significance for multi-group curve comparisons; Log-rank test was used to assess statistical significance for survival curve comparisons; unless otherwise noted. The $p$ values are indicated in the plots. Source data and additional statistics for experiments are provided as a Source Data file.

(pPR023), containing cDNA sequence of the constant region of light chain, CH2 and CH3 with "hole" mutations, and regions overlapping with corresponding flanking sequences of NsiI and NheI restriction sites in pFUSEss-CHIg-hG1 was ordered from IDT and cloned into NsiI and NheI restriction enzymes digested pFUSEss-CHIg-hG1 backbone through Gibson assembly (NEB). All plasmids were sequenced and Maxiprepped for subsequent experiments.

A list of oligos used for plasmid construction is provided in Supplemental Table 1 in the Supplemental Information.

**Cloning of SARS-CoV-2 spike variants.** The construct of wild-type (WT) SARS-CoV-2 ectodomain of spike trimer is a gift from Dr. Jason S. McLellan at the University of Texas at Austin[34]. The recently emerged SARS-CoV-2 spike SA variant B.1.351[7] and Indian variant B.1.617[11] was generated by standard cloning. The pVP21-SA variant includes four mutations in the N-terminal domain (L18F, D80A and D215G, R246I), three mutations at key residues in the RBD (N501Y, E484K, and K417N), and one is in loop 2 (A701V). The pVP28-Indian variant includes seven mutations in Spike G142D, E154K, L452R, E484Q, D614G, P681R, and Q1071H. The pVP21-SA and pVP28-Indian were generated based on pcDNA3.1-pSARS-CoV-2-S, which was derived from a synthetic human codon-optimized cDNA (Geneart) encoding a WA1 SARS-CoV-2 S protein. For pVP21-SA-variant, two gBlocks, contain mutations in SA variant regions overlapping with corresponding flanking sequences of NheI and BsrGI restriction sites pcDNA3.1-pSARA-CoV-2. The gBlocks were then cloned into the pcDNA3.1-pSARA-CoV-2 backbone, digested with NheI and BsrGI restriction enzyme (Thermo Fisher) through Gibson assembly. For pVP28-Indian, four gBlocks, contains mutations in Indian variant regions overlapping with corresponding flanking sequences of NheI and BamHI restriction sites pcDNA3.1-pSARA-CoV-2. The gBlocks were then cloned into the pcDNA3.1-pSARA-CoV-2 backbone, digested with NheI and BamHI restriction enzyme (Thermo Fisher) through Gibson assembly. For the HIV-1-based SARS-CoV-2 spike pseudotyped virus generation, WT pcDNA3.1-pSARS-CoV-2-S, pVP21-SA-variant, and a pVP28-Indian variant lacking the C-terminal 19 codons were employed. A pair of forward and reverse primers were utilized to amplify fragments lacking the C-terminal 19 codons with the pVP21-SA variant and pVP28-Indian variant as templates separately. The amplified fragments were gel-purified and cloned into the pVP21-SA variant backbone and pVP28-Indian variant backbone, digested with BbvCI and BamHI.

**Cell culture.** HEK293FT (Thermo Fisher) and 293T-hACE2 (gifted from Dr. Bieniasz' lab) cell lines were cultured in the complete growth medium, Dulbecco's modified Eagle's medium (DMEM; Thermo Fisher) supplemented with 10% Fetal bovine serum (FBS, Hyclone),1% penicillin-streptomycin (Gibco) (D10 media for short). Cells were typically passaged every 1–2 days at a split ratio of 1:2 or 1:4 when the confluency reached 80%. Expi293F$^{TM}$ (Thermo Fisher) cells were cultured in Expi293$^{TM}$ Expression Medium (Thermo Fisher) in a 125-mL shaker flask in a 37 °C incubator with 8% $CO_2$ on an orbital shaker rotating at 125 rpm. For routine maintenance, Expi293F$^{TM}$ cells were grown to $3–5 \times 10^6$ cells/mL, then split to $0.3–0.5 \times 10^6$ cells/mL every 3 days.

**Expression and purification of WT SARS-CoV-2 ectodomain of spike trimer.** The WT ectodomain of the SARS-CoV-2 spike trimer was expressed in Expi293F cells. For 100 mL expression scale, 100 μg construct DNA was mixed with 400 μg polyethylenimine in 10 mL Opti-MEM$^®$ I Reduced-Serum Medium (Thermo Fisher) for 30 min, and then added into 90 mL Expi293F cells at a density of $2.5–3 \times 10^6$ cells/mL for incubation, shaking at 125 rpm in a 37 °C incubator with 8% $CO_2$. After 5 days, the medium with the secreted protein was harvested and

loaded onto an ion-exchange column. Fractions containing the target protein was pooled and further purified using a Ni-NTA affinity column, followed by size exclusion chromatography using a Superose 6 10/300 column (GE Healthcare) with a buffer of 30 mM Tris, pH 8.0, 100 mM NaCl. The monodispersed peak containing the ectodomain of the spike trimer was pooled and concentrated for subsequent analysis.

**Recombinant antibody generation.** The top-ranked enriched IgG clones were selected and cDNAs of a relative variable region of paired heavy- and light-chain were codon-optimized and cloned separately into human IgG1 heavy chain and light chain expression vectors, containing the human IgG1 constant regions (pFuse plasmids). IgG1 antibodies were expressed in Expi293F$^{TM}$ cells. ExpiFectamine 293 transfection kit (Thermo Fisher) was utilized for heavy and light chain plasmids transfection following the manufacturer's instruction. After 5 days, the antibody-containing supernatants were collected. A suitable amount of rProtein A Sepharose$^®$ Fast Flow beads (Cytiva) was prewashed and added into supernatants. After overnight incubation at 4 °C, antibody-bound protein A beads were collected with Poly-Prep$^®$ Chromatography Columns (BIO-RAD). After three times wash with DPBS, mAbs were eluted with Fab elution buffer, then neutralized with Tris-HCl. Buffer exchange was performed with Amicon Ultra-4 Centrifugal Filter (MilliporeSigma) to keep mAbs in PBS for the following assays. The numbering of mAbs was based on the order of mouse immunization and cloning. Clones 1–4 were mAbs chosen from enriched clones from RBD-his tag protein immunized C57BL/6 J mice. Clone 5-11 were mAbs chosen from RBD-his tag protein immunized BALB/c mice.

**Bispecific antibody generation.** Clone 16 (Clone 6-KiH-Clone 2) bispecific antibody is a human IgG1-like bispecific antibody, generated based on CrossMab-KiH bispecific constructs, including pFUSE2ss-knobLight-hK, pFUSE2ss-knob-heavy-hG1, pFUSE2ss-HoleLight-hK, and pFUSE2ss-HoleHeavy-hG1. The design and generation of CrossMab-KiH bispecific constructs was described in the above plasmid constructs parts. The variable region of the Clone 6 heavy chain was cloned into pFUSE2ss-knobheavy-hG1 vector. The variable region of the Clone 6 light chain was cloned into pFUSE2ss-knobLight-hK vector. Clone 6-KiH-Clone 2 bispecific antibody was expressed in vitro in Expi293F$^{TM}$ cells by co-transfecting four plasmids (Clone 6 knob heavy chain plasmid, Clone 6 knob light chain plasmid, Clone 2 hole heavy chain plasmid, and Clone 2 hole light chain plasmid) with ExpiFectamine 293 transfection kit (Thermo Fisher). The expression and antibody purification protocol was similar to the recombinant antibody expression described above. The bispecific antibody was efficiently purified by using rProtein A Sepharose Fast Flow antibody purification resin (Cytiva, Cat:#17127901).

**Antibody humanization.** In order to humanize the antibody, we first determine the six CDR loops from murine variable domains by using the online free program "IGBLAST" (https://www.ncbi.nlm.nih.gov/igblast/). Followed by applying the CDR-grafting technique and grafting six CDR loops onto human acceptor frameworks. The framework template selection was based on sequence similarity to close human germline sequence, as well as homology to clinically validated germline sequences. Thereafter, we identify Vernier zone residues through Cryo-EM structure between Clone 2 and trimeric S protein of SARS-CoV-2 from parent antibody (FR residues of Clone 2 within 5 Å of trimeric S protein) and substitute the key residues into the human acceptor framework of Clone 13 A.

**ELISA**

*ELISA for anti-serum titer determination.* The antibody titers in sera from pre, first, and second bleeds were determined using direct coating ELISA. The 384-well

ELISA plates (Corning) were coated with 3 μg/mL SARS-CoV-2 RBD-his tag protein (Sino) in PBS at 4 °C overnight. After standard washing with PBST washing buffer (phosphate-buffered saline containing 0.05% Tween 20), ELISA plates were blocked with blocking buffer (2% bovine serum albumin dissolved in PBST and filtered) for 1 h at room temperature. Serial dilutions of pre-immune, first, and second immune anti-sera in blocking buffer were added into plates and for 1 h at room temperature. Plates were washed and incubated with relative goat anti-mouse IgG(H + L)/HRP (Thermo Fisher,1:5000) for 1 h at room temperature. Plates were washed and developed using TMB reagents as substrates (Biolegend) following the manufacturer's recommended protocol. The reaction was stopped with a stop solution (1 M $H_3PO_4$) and absorbance at 450 nm was recorded by a microplate reader (Perkin Elmer).

*ELISA for Anti-RBD antibody clones binding.* SARS-CoV-2 RBD protein with 6 X Histidine (Sino) was coated at 3 μg/ml in PBS on a 384-well microtiter plate overnight at 4 °C. After standard washing with PBST and blocked with 2% (w/v) solution of BSA in PBST to remove the nonspecific binding, purified anti-RBD antibodies were diluted proportionally in PBST + 2% BSA and transferred to the washed and blocked microtiter plates. After 1 h of incubation at RT, plates were washed, and RBD-his tag protein-bound antibody was detected with goat anti-human IgG1 (H + L) with horseradish peroxidase (HRP) conjugated (Invitrogen,1:1000) The plates were washed and developed using TMB substrate solution (Biolegend) according to manufacturer's recommendation and absorbance at 450 nm was measured on a microplate reader after the reaction was stopped by stop solution (1 M $H_3PO_4$).

**Affinity determination via bio-layer interferometry (BLI).** Antibody binding kinetics for anti-spike mAbs were evaluated by BLI on an Octet RED96e instrument (FortéBio) at room temperature. Two types of measurements were performed. (1) HIS1K biosensors (FortéBio) were first loaded with his-tagged SARS-CoV-2 RBD protein to a response of about 1 nm, followed by a 60 s baseline step in the kinetic buffer (PBS, 0.02% Tween, pH 7.4). After that, the biosensors were associated with indicated concentrations of the antibodies (from 50 to 0.78125 nM with twofold dilutions, where the kinetic buffer was served as the negative control) for 200 s, then dissociated in the kinetic buffer for 1000 s. (2) 25 ng/ul of Clone 13A-IgG1 antibodies were captured on an AHC biosensor (ForteBio). The baseline was recorded for 60 s in a running buffer (PBS, 0.02% Tween 20, and 0.05% BSA, pH 7.4). Afterward, the sensors were subjected to an association phase for 500 s in wells containing RBD-his diluted in the buffer. In the dissociation step, the sensors were immersed in the running buffer for 1000 s. The dissociation constants KD, kinetic constants $K_{on}$ and $K_{off}$, were calculated by using a 1:1 Langmuir binding model with FortéBio data analysis software. Octet data were analyzed with Octet® CFR software and Prism.

**Affinity measurement by surface plasmon resonance (SPR).** Kinetics binding measurement for anti-spike mAbs in this study was performed using a Biacore T200 instrument (GE Healthcare). The system was flushed with filtered 1xHBS-P + running buffer (0.01 M HEPES, 0.15 M NaCl, and 0.05%v/v Surfactant P20, pH 7.4) and all steps were performed at 25 °C chip temperature.

*Kinetics binding measurement on CM5 Chip (Series S sensor chip CM5).* For kinetic binding measurements, the CM5 chip surface was activated by injecting a solution of EDC/NHS (GE Healthcare). Mouse anti-human IgG (Fc) mAb (25 μg/ml) was immobilized on the sensor chip by amine coupling, followed by deactivation using 1 M ethanolamine. Afterward, anti-spike mAbs (0.1 μg/ml) were then flowed over and captured on anti-human IgG (Fc) mAb-coated surface. Subsequently, gradient diluted his-tagged SARS-CoV-2 RBD solutions (1.875–30 nM, twofold serial dilution) were injected individually in a single-cycle kinetic format without regeneration (30 μl/min, association:180 s, dissociation:60 s). The binding data were double referenced by blank cycle and reference flow cell subtraction. Processed data were fitted by a 1:1 interaction model using Biacore T200 Evaluation Software 3.1.

*Kinetics binding measurement on NTA Chip.* For kinetic binding measurements, the NTA chip was activated manually by loading a solution of NiCl2. Histidine-labeled SARS-CoV-2 RBD protein (0.075 μg/ml) was then flowed over the chip and captured on a nickel-coated surface. Subsequently, gradient diluted anti-spike mAbs solutions (0.9875–15 nM, twofold serial dilution) were injected individually in a single-cycle kinetic format without regeneration (30 μl/min, association:240 s, dissociation:90 s). The binding data were double referenced by blank cycle and reference flow cell subtraction. Processed data were fitted by a 1:1 interaction model using Biacore T200 Evaluation Software 3.1.

**SARS-CoV-2 pseudovirus reporter and neutralization assays.** HIV-1 based SARS-CoV-2 S pseudotyped virions were generated according to a previous study[70]. Two plasmids are adopted to generate HIV-1 based SARS-CoV-2 S pseudotyped virions. HIV-1 dual reporter vector expressing mCherry and luciferase (NL4-3 mCherry Luciferase, plasmid#44965) was purchased from Addgene. Plasmid expression of a C-terminally truncated SARS-CoV-2 S protein (pSARS-CoV-2Δ19) was obtained from Dr. Bieniasz's lab. In order to generate HIV-1 based

SARS-CoV-2 S pseudotyped virions, 15×$10^6$ 293FT cells were seeded in 150 mm plates one day before in 20 ml D10 media. The following day, after the cell density reaches 90%, the medium was discarded and replaced with a 13 mL serum-free Opti-MEM medium. 20 μg NL4-3 mCherry Luciferase reporter plasmids and 15 μg SARS-CoV-2 (pSARS-CoV-2Δ19) plasmids were mixed thoroughly in 225 μl serum-free Opti-MEM medium. Then 100 μl Lipofectamine 2000 (Invitrogen) were diluted in 225 μl serum-free Opti-MEM medium. Then the diluted plasmid mixture and Lipofectamine 2000 were mixed thoroughly and incubated for 10 min at RT before adding into cells. After 6 h, the culture medium was changed back to the completed growth medium, 20 mL for one 150 mm plate. At 48 h after transfection, the 20 mL supernatant was harvested and filtered through a 0.45-μm filter, aliquoted, and frozen in −80 °C.

Parallelly, the three plasmids-based HIV-1 pseudotyped virus systems were utilized to generate (HIV-1/NanoLuc2AEGFP)-SARS-CoV-2 particles and (HIV-1/NanoLuc2AEGFP)-SARS-CoV-2-SA variant particles. The reporter vector, pCCNanoLuc2AEGFP, and HIV-1 structural/regulatory proteins (pHIVNLGagPol) expression plasmid were gifts from Dr. Bieniasz's lab[70]. Briefly, 293 T cells were seeded in 150 mm plates and transfected with 21 μg pHIVNLGagPol, 21 μg pCCNanoLuc2AEGFP, and 7.5 μg of a SARS-CoV-2 SΔ19 or SARS-CoV-2 SA SΔ19 plasmid utilizing 198 μl PEI. At 48 h after transfection, the 20-ml supernatant was harvested and filtered through a 0.45-μm filter, and concentrated before aliquoted and frozen at −80 °C.

The pseudovirus neutralization assays were performed on 293T-hACE2 cell line[70]. One day before, 293T-hACE2 cells were plated in a 96-well plate, 0.02 × $10^6$ cells per well. The following day, serial dilution of monoclonal IgG from 40 μg/mL (fourfold serial dilution using complete growth medium, 55 μL aliquots) were mixed with the same volume of SARS-CoV-2 pseudovirus. The mixture was incubated for 1 h at 37 °C incubators, supplied with 5% CO2. Then 100 μL of the mixtures were added into 96-well plates with 293T-hACE2 cells. Plates were incubated at 37 °C supplied with 5% $CO_2$. Forty-eight hours later, 1 μL D-luciferin reagent (Perkin Elmer, 33.3 mg/ml) was added to each well and incubated for 5 min. Luciferase activity was measured using a microplate spectrophotometer (Perkin Elmer). The inhibition rate was calculated by comparing the OD value to relative negative and positive control wells. For the three plasmids-based HIV-1 pseudotyped virus systems, 293 T cells were collected and the GFP + cells were analyzed with Attune NxT Acoustic Focusing Cytometer (Thermo Fisher). The 50% inhibitory concentration (IC50) was calculated with a four-parameter logistic regression using GraphPad Prism 8.0 (GraphPad Software Inc.).

**Cell fusion assay**

*Vectors and plasmids.* Plasmid encoding human ACE2 (hACE2) was obtained from Addgene (hACE2; catalog #1786). The hACE2 2.6 kbp ORF was also blunt-cloned into a third-generation HIV vector 3′ of the CMV promoter and 5′ of an IRES-puro$^r$ cassette to generate pHIV-CMV-hACE2-IRES-Puro. It was inserted into a *piggybac* transposon (Matt Wilson of Baylor College of Medicine, along with the transposase plasmid pCMV-*piggybac*) that had been modified to encode a CMV-IRES-*bsd*$^r$ cassette; the resultant plasmid was named pT-PB-SARS-CoV-2 Spike-IRES-Blasti. This too was inserted into *piggybac* transposon to make pT-PB-SARS-CoV-2-UK Spike-IRES-Blasti.

*Cell lines.* The HOS cells were stably transduced with a third-generation HIV vector encoding *tat*, along with eGFP, mRFP, and bleomycin resistance gene; they were maintained in 200–400 μg/mL phleomycin (Invivogen) and were eGFP and mRFP-positive by flow cytometry. hACE2 was subsequently introduced by VSV G-mediated HIV-based transduction using pHIV-CMV-hACE2-IRES-Puro to produce HOS-3734, which cell lines maintained in selection using 10 μg/mL puromycin (Sigma-Aldrich). TZMbl cells (#JC53BL-13) were obtained from the NIH AIDS Reagent Program. TZMbl cells stably expressing wild-type S/UK variant S were created by co-transfecting TZMbl cells with pT-PB-SARS-CoV-2- Spike-IRES-Blasti or pT-PB-SARS-CoV-2-UK Spike-IRES-Blasti, respectively, along with pCMV-*piggybac* and resistant cells selected with 10 μg/mL blasticidin (Invivogen). The control TZMbl cell line not expressing S was generated by co-transfecting pCMV-*piggybac* with pT-pB-IRES-Blasti and selecting for blasticidin-resistant TZMbl cells.

*Cell fusion inhibition by monoclonal antibodies.* Producer cells (TZMbl-wild-type Spike/ Tzmbl-UK Spike) and target cells (HOS-3734) were generated as described above. Ten thousand S-expressing cells (TZMbl-wild-type Spike/TZMbl-UK Spike) in 100 μL of the medium in the absence of blasticidin were seeded in 96-well plates. After 24 h, 70 μL of fourfold serially diluted antibody was added into producer cells and incubated at 37 °C for 1 h. At that time $10^4$ target cells (HOS-3734) in 50 μL medium were then added to the producer cells, and after another 24 h cells were lysed in 0.1 mL and RLU measured. Data were analyzed with nonlinear regression using GraphPad Prism to determine the neutralization curve and the IC$_{50}$ values calculated.

**In vitro neutralization against authentic SARS-CoV-2.** SARS-CoV-1 (USA-WA1/2020) was produced in Vero-E6 cells and tittered as described previously[71]. SARS-CoV-2 neutralization was assessed by measuring cytotoxicity. About 5 × $10^5$

Vero-E6 cells were plated per well of a 96-well plate. The following day, serial dilutions of antibodies were incubated with $2.5 \times 10^3$ plaque-forming units (PFU) SARS-CoV-2 for 1 h at room temperature. SARS-CoV-2 neutralization was assessed by measuring cytotoxicity. About $5 \times 10^5$ Vero-E6 cells were plated per well of a 96-well plate. The following day, serial dilutions of antibodies were incubated with $2.5 \times 10^3$ PFU SARS-CoV-2 for 1 h at room temperature. The medium was then aspirated from the cells and replaced with 100 μl of the antibody/virus mixture. After 72 h at 37 °C, 10 μl of CellTiter- Glo (Promega) was added per well to measure cellular ATP concentrations. Relative luminescence units were detected on Cytation5 (Biotek) plate reader. All conditions were normalized to uninfected control. Each condition was done in triplicate in each of three independent experiments.

**Focus reduction neutralization test**. Serial dilutions of mAbs or sera were incubated with $10^2$ focus-forming units (FFU) of different strains or variants of SARS-CoV-2 for 1 h at 37 °C. Antibody-virus complexes were added to Vero-TMPRSS2 cell monolayers in 96-well plates and incubated at 37 °C for 1 h. Subsequently, cells were overlaid with 1% (w/v) methylcellulose in MEM supplemented with 2% FBS. Plates were harvested 24 h later by removing overlays and fixed with 4% PFA in PBS for 20 min at room temperature. Plates were washed and sequentially incubated with an oligoclonal pool of SARS2-2, SARS2-11, SARS2-16, SARS2-31, SARS2-38, SARS2-57, and SARS2-71[4,72]. The anti-S antibodies and HRP-conjugated goat anti-mouse IgG (Sigma, 12-349) in PBS supplemented with 0.1% saponin and 0.1% bovine serum albumin. SARS-CoV-2-infected cell foci were visualized using TrueBlue peroxidase substrate (KPL) and quantitated on an ImmunoSpot microanalyzer (Cellular Technologies).

**In vivo efficacy testing against authentic SARS-CoV-2**. The efficacy of mAbs against replication-competent SARS-CoV-2 virus was evaluated in vivo, using both a prophylactic setting where the animals were treated with mAb prior to viral infection and a therapeutic setting where the animals were treated post-infection. These experiments were performed in an animal BSL3 (ABSL3) facility. The replication-competent SARS-CoV-2 (USA-WA1/2020) virus was produced in Vero-E6 cells, and the titer was determined by plaque assay using WT Vero-E6.

The K18-hACE2 mice (B6.Cg-Tg(K18-ACE2)2Prlmn/J) were purchased from the Jackson Laboratory and bred in-house using a trio breeding scheme. Mice were sedated with isoflurane and infected via intranasal inoculation of 2000 PFU (20x LD50) SARS-CoV-2 (USA-WA1/2020) virus administered in 50 uL of DPBS. Six to eight-week-old K18-hACE2 littermate-controlled mice, mixed-gender (male/female) mice were divided randomly into three groups and administered with 20 mg/kg (of mice body weight) Clone 2, Clone 6 or placebo/control, via intraperitoneal (IP) injection. For a prophylactic experiment, the mAb drug/placebo treatment was 24 h prior to the infection; for a therapeutic experiment, the treatment was 18 h post-infection. The control for the prophylactic experiment was DPBS, and the control for the therapeutic experiment was isotype control hIgG1, where both controls are similar (no effect on disease progression). Survival, body conditions, and weights of mice were monitored daily for 10 consecutive days.

**In vivo efficacy testing of humanized Clone 13A to authentic SARS-CoV-2 virus**. Ten-12-week-old littermate-controlled female and male K18hAce2Tg+ mice were pretreated with 20 mg/kg of either control hIgG1 (purchased from BioXCell) or clone 13A mAb (produced by the Chen lab) administered IP in 300 uL of DPBS. Twenty-four hours later, mice were anesthetized with isoflurane, and SARS-CoV-2 isolate USA-WA1/2020, or Delta variant (B.1.617.2), was inoculated intranasally at a dose of $2 \times 10^3$ PFU/mouse (determined using wild-type Vero-E6) in 50 μL of DPBS. Weights were obtained daily for 10 days following infection, and mice were euthanized when morbid.

**Fab generation**. The Fab fragments of Clone 2 and Clone 6 were generated from full-length IgGs of Clone 2 and Clone 6 using a commercial PierceTM Fab Preparation Kit (Thermo Fisher). All procedures were performed following the manufacturer's instructions. Briefly, 2 mg of the whole IgGs of Clone 2 and Clone 6 were digested with immobilized papain at 37 °C for 4 h with rotation. Then protein A beads were applied to bind the Fc fragments and undigested IgG. Then Fab fragments were recovered in the flow-through fraction, and further purified by size exclusion chromatography using a Superdex 200 10/300 column (GE Healthcare) in 30 mM Tris pH 8.0, 100 mM NaCl. The monodispersed peak of Fab fragments was pooled and concentrated for subsequent analysis.

**Cryo-EM sample preparation and data collection**. The purified SARS-CoV-2 spike trimer at a final concentration of 0.3 mg/mL (after mixture) was mixed with Clone 2 or Clone 6 Fab at a molar ratio of 1:2 at 4 °C for 30 min. Then 3 μl of the protein mixture was applied to a Quantifoil-Cu-2/1-3 C grid (Quantifoil) pretreated by glow-discharging at 15 mA for 1 min. The grid was blotted at 4 °C with 100% humidity and plunge-frozen in liquid ethane using FEI Vitrobot Mark IV (Thermo Fisher). The grids were stored in liquid nitrogen until data collection.

Images were acquired on an FEI Titan Krios electron microscope (Thermo Fisher) equipped with a Gatan K3 Summit direct detector in super-resolution mode, at a calibrated magnification of 81,000× with the physical pixel size

corresponding to 1.068 Å. Detailed data collection statistics for the Fab-spike trimer complexes are shown in a Supplemental Table. Automated data collection was performed using SerialEM 3.8[73].

**Cryo-EM data processing**. A total of 2655 and 1766 movie series were collected for Clone 2 Fab-S trimer complex and Clone 6 Fab-S trimer complex, respectively. The same data processing procedures were carried out for each complex as described below. Motion correction of the micrographs was carried out using RELION[74] and contrast transfer function (CTF) estimation was calculated using CTFFIND4[75]. Particles were picked automatically by crYOLO[76], followed by 2D and 3D classifications without imposing symmetry. The 3D classes with different S trimer conformations were then processed separately by consensus 3D refinement and CTF refinement. Image processing and 3D reconstruction using cryoSPARC[77] produced similar results. For each state of the Clone 6 Fab-S trimer complex, multibody refinements were then carried out in RELION by dividing the complex into individual rigid bodies (three refinements each with a rigid body containing a unique Fab, RBD, and the N-terminal domain (NTD) of spike S1 subunit, and another rigid body for the rest of the spike-ectodomain trimer). For each state of the Clone 2 Fab-S trimer complex, local masked 3D classification without image alignment was performed focusing on one Fab-RBD region, and the best class of particles was selected for consensus refinement of the whole complex. Subsequently, multibody refinement was performed as described above for the rigid body containing the focused region. The 3D reconstruction of the other Fab-RBD regions were obtained with the same procedure. The final resolution of each reconstruction was determined based on the Fourier shell correlation (FSC) cutoff at 0.143 between the two half maps[78]. The final map of each body was corrected for K3 detector modulation and sharpened by a negative B-factor estimated by RELION[79], and then merged in Chimera for deposition. The local resolution estimation of each cryo-EM map is calculated by RELION[74]. See also Supplementary Fig. 7 and Table 1.

**Model building and refinement**. The structure of the ectodomain of SARS-CoV-2 spike trimer (PDB 6VSB) was used as an initial model and docked into the spike trimer portion of the cryo-EM maps using Chimera[80]. The initial models of Clone 2 and Clone 6 Fabs were generated by homology modeling using SWISS-MODEL[81], and then docked into the Fab portions of the cryo-EM maps using Chimera[80]. The initial models were subsequently manually rebuilt in COOT[82], followed by iterative cycles of refinement in Refmac[83] and PHENIX[84]. The final models with good geometry and fit to the map were validated using the comprehensive cryo-EM validation tool implemented in PHENIX[85]. All structural figures were generated using PyMol (http://www.pymol.org/) and ChimeraX[80].

**Homology modeling of SARS-CoV-2 variants**. The structural models of SARS-CoV-2 variants of RBD were generated by SWISS model[81] using the wildtype / WA RBD Cryo-EM structure as a template. The generated structures were aligned with the wild-type RBD in complex with Clone 2, Clone 6, and/or other mAbs. The cryo-EM structures and homology models were analyzed in Pymol.

**Replication, randomization, blinding, and reagent validations**. Sample size determination was performed according to similar work in the field, e.g., (Wang et al. 2021 Nature).

Replicate experiments have been performed for key data shown in this study, as detailed in methods and/or legends. Replicate experiments were successful where applicable.

Biological or technical replicate samples were randomized where appropriate. In animal experiments, mice were randomized by cage, sex, and littermates.

Experiments were not blinded. It is unnecessary for animal immunization for antibody production to be blinded. Cryo-EM structure study can not be blinded.

**Antibodies and dilutions**. Commercial antibodies used for staining were the following, with typical dilutions listed:

Mouse anti-Human IgGl Fe Secondary Antibody, HRP Thermo Fisher Cat#A-10648, 1:2000

InVivoMAb human IgGl isotype control BioXcell Cat#BE0297.

Goat Anti-Mouse IgG H&L (HRP) Abcam ab6789, Abcam Cat#ab6789, 1:5000.

Recombinant monoclonal human IgGl antibody against Spike RBD Invivogen Cat#srbd-mabl

Custom antibodies were generated in this study, where dilutions were often serial titrations (i.e., a number of dilutions as specified in each figure)

Anti-SARS-CoV-2 Spike mAbs:

Clone 1
Clone 2
Clone 3
Clone 4
Clone 5
Clone 6
Clone 7
Clone 8
Clone 9

Clone 11
Clone 12
Clone 13
Clone 13A
Clone 16 (bispecific)

Commercial antibodies were validated by the vendors, and re-validated in-house as appropriate. Custom antibodies were validated by specific antibody–antigen interaction assays, such as ELISA. Isotype controls were used for antibody validations.

Commercial antibody info and validation info where applicable:
https://www.thermofisher.com/antibody/product/Mouse-anti-Human-IgG1-Fc-Secondary-Antibody-clone-HP6069-Monoclonal/A-10648
https://bxcell.com/product/invivomab-human-igg1-isotype-control/
https://www.abcam.com/goat-mouse-igg-hl-hrp-ab6789.html

**Eukaryotic cell lines**. Cell line sources: Various, e.g., HEK293FT, Thermo Fisher Cat#R70007

HEK293T-hACW2, Dr Bieniasz's lab
Vero-E6, ATCC, Cat#CRL-1586™
Expi293F™, Thermo Fisher Cat#A14527
HOS-3734, ATCC, Cat#CRL-1543™
TZMbl, Dr. Sutton's lab.

Cell lines were authenticated by original vendors, and re-validated in the lab as appropriate, by morphology and PCRs.

All cell lines tested negative for mycoplasma.

No commonly misidentified lines involved.

**Animals and other organisms**. Laboratory animals: *M. musculus*,
C57BL/6 J, Jackson laboratory, Cat#000664
B6.Cg-Tg(K18-ACE2)2Prlmn/J, Jackson laboratory, Cat#034860
BALB/c, Jackson laboratory, Cat#000651

Animals are maintained and bred in standard individualized cages with a maximum of five mice per cage, at regular room temperature (65–75 °F, or 18–23 °C), 40–60% humidity, and a 12 h:12 h light cycle for breeding, and 13 h:11 h or 14 h:10 h light cycle for experiments.

Wild animals: No wild animals were used in this study.

Field-collected samples: No field-collected samples were used in this study.

**Software and codes**

*Data collection*. ELISA data were recorded by a microplate reader (Perkin Elmer) (no version number).

Antibody binding kinetics for anti-spike mAbs were evaluated by BLI on an Octet RED96e instrument (FortéBio) at room temperature (version 12).

Affinity measurement by surface plasmon resonance (SPR): Kinetics binding measurement for anti-spike mAbs in this study was performed using a Biacore T200 instrument (GE Healthcare) (v3).

Cryo-EM data were acquired on an FEI Titan Krios electron microscope (Thermo Fisher) equipped with a Gatan K3 Summit direct detector in super-resolution mode, at a calibrated magnification of 81,000× with the physical pixel size corresponding to 1.068 Å. Detailed data collection statistics for the Fab-spike trimer complexes are shown in a Supplemental Table. Automated data collection was performed using SerialEM (v3.8).

*Data analysis*. Standard biological assays' data were analyzed in Prism (v8 or v9).

SPR results were analyzed by using Biacore T200 Evaluation Software 3.0.

BLI data were analyzed by using Octet Analysis Studio Software 10.0.

Motion correction of the micrographs was carried out using RELION (v3.1.2) and contrast transfer function (CTF) estimation was calculated using CTFFIND4 (v4.1).

Particles were picked automatically by crYOLO (v1.8.1), followed by 2D and 3D classifications without imposing symmetry. The 3D classes with different S trimer conformations were then processed separately by consensus 3D refinement and CTF refinement.

Image processing and 3D reconstruction using cryoSPARC (v3.3.1) produced similar results.

The final map of each body was corrected for K3 detector modulation and sharpened by a negative B-factor estimated by RELION and then merged in Chimera for deposition.

The structure of the ectodomain of the SARS-CoV-2 spike trimer (PDB 6VSB) was used as an initial model and docked into the spike trimer portion of the cryo-EM maps using Chimera (v1.15).

The initial models of Clone 2 and Clone 6 Fabs were generated by homology modeling using SWISS-MODEL (online https://swissmodel.expasy.org) and then docked into the Fab portions of the cryo-EM maps using Chimera.

The initial models were subsequently manually rebuilt in COOT (0.9.7), followed by iterative cycles of refinement in Refmac5 (v5) and PHENIX (v1.19).

The final models with good geometry and fit to the map were validated using the comprehensive cryo-EM validation tool implemented in PHENIX.

All structural figures were generated using PyMol (v1.3) (online http://www.pymol.org/) and ChimeraX (v1.2).

**Reporting summary**. Further information on research design is available in the Nature Research Reporting Summary linked to this article.

## Data availability

All data generated or analyzed during this study are included in this article, source data, and Supplementary Information Files. Specifically, source data and statistics for non-high-throughput experiments are provided in a Supplementary Table excel file, Source data are provided with this paper. High-throughput experiment data are provided as processed quantifications in Supplemental Datasets. Genomic sequencing raw data have been deposited to Gene Expression Omnibus (GEO) with the accession code (GSE174635), The models of the mAb:Spike complexes have been deposited in the wwPDB with accession codes: 3dSpike-Fab6,7MW2; 2dSpike-Fab6, 7MW3; 1dSpike-Fab6, 7MW4; 2uSpike-Fab2, 7MW5; 3uSpike-Fab2, 7MW6. The cryo-EM maps of the mAb:Spike complexes have been deposited in EMDB with accession codes: 3dSpike-Fab6, EMD-24060; 2dSpike-Fab6, EMD-24061; 1dSpike-Fab6, EMD-24062; 2uSpike-Fab2, EMD-24063; 3uSpike-Fab2, EMD-24064. Additional information related to this study are available from the corresponding authors upon reasonable request. Source data are provided with this paper.

## Code availability

Codes that support the findings of this research are implementation of software publicly available, as noted in the methods section.

Cellranger is available at (https://support.10xgenomics.com/single-cell-gene-expression/software/pipelines/latest/what-is-cell-ranger). Loupe Browser is available at (https://www.10xgenomics.com/products/loupe-browser).

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

## Acknowledgements

We thank various members from Chen, Xiong, and Wilen labs for discussions and support. We thank Dr. Bieniasz for providing pseudovirus reporter plasmids. We thank the staff from various Yale core facilities (Keck, YCGA, HPC, Biophysics, YARC, Cryo-EM, CBDS, and others) for technical support. We thank various support from Departments of Genetics, MBB, Laboratory Medicine, Immunobiology, Internal Medicine and Pharmacology; Institutes of Systems Biology and Cancer Biology; Dean's office of Yale School of Medicine and the office of Vice Provost for Research. This work is supported by DoD PRMRP IIAR (W81XWH-21-1-0019) and discretionary funds to S.C.; discretionary funds to Y.X.; Ludwig Foundation, Mathers Foundation, Burroughs Wellcome Fund, NIH K08 AI128043, NIH R01 AI148467 to C.B.W.; NIH R01 AI157155 to M.S.D.; NIH/NIAID R01 AI150334 to R.S. YCGA / HPC were supported by NIH Award 1S10OD018521. The T200 Biacore instrumentation was supported by NIH Award S10RR026992-0110.

## Author contributions

S.C. conceived the study. L.P. and P.R. immunized the animals, generated the mAb clones, and characterized their properties. Y.H. led the solution of Cryo-EM structures. M.C.M. performed BSL3 in vivo experiments. R.E.C. and J.W. performed BSL3 in vitro experiments. M.Z. performed a cell fusion assay. T.L., L.Y., M.B.D., and P.C. assisted various experiments. T.T., C.W., and M.C. assisted Cryo-EM experiments and analyses. R.D.C., Z.F., and G.W. assisted certain data analyses. B.N. and D.K. provided support on biophysical analysis. R.S. provided support on fusion and neutralization assays. M.S.D. provided support on BSL3 in vitro experiments. D.K., R.S., M.S.D., C.B.W., Y.X., and S.C. secured funding. S.C., C.B.W., and Y.X. jointly supervised the work. L.P., Y.H., M.C.M., P.R., R.E.C., R.S., M.S.D., C.B.W., Y.X., and S.C. prepared the manuscript with inputs from all authors.

## Competing interests

A patent application has been filed by Yale University on the antibodies described here (inventors: S.C., L.P., and P.R.). Yale University has committed to rapidly executable nonexclusive royalty-free licenses to intellectual property rights for the purpose of making and distributing products to prevent, diagnose, and treat COVID-19 infection during the pandemic and for a short period thereafter. S.C. is a co-founder of EvolveImmune Tx and Cellinfinity Bio, unrelated to the study. M.S.D. is a consultant for Inbios, Vir Biotechnology, Senda Biosciences, and Carnival Corporation, and on the Scientific Advisory Boards of Moderna and Immunome. The Diamond laboratory has received unrelated funding support in sponsored research agreements from Vir Biotechnology, Moderna, and Emergent BioSolutions. The remaining authors declare no competing interests.

## Additional information

[1]Department of Genetics, Yale University School of Medicine, New Haven, CT, USA. [2]System Biology Institute, Yale University, West Haven, CT, USA. [3]Center for Cancer Systems Biology, Yale University, West Haven, CT, USA. [4]Department of Molecular Biophysics and Biochemistry, Yale University, New Haven, CT, USA. [5]Department of Laboratory Medicine, Yale University, New Haven, CT, USA. [6]Department of Immunobiology, Yale University, New Haven, CT, USA. [7]Departments of Medicine and Pathology & Immunology, Washington University School of Medicine in St. Louis, St. Louis, MO, USA. [8]Section of Infectious Diseases, Department of Internal Medicine, Yale University, New Haven, CT, USA. [9]Department of Pharmacology, Yale University School of Medicine, New Haven, CT, USA. [10]Cancer Biology Institute, Yale University, West Haven, CT, USA. [11]M.D.-Ph.D. Program, Yale University, West Haven, CT, USA. [12]Molecular Cell Biology, Genetics, and Development Program, Yale University, New Haven, CT, USA. [13]Immunobiology Program, Yale University, New Haven, CT, USA. [14]Yale Center for Genome Analysis, Yale University, New Haven, CT, USA. [15]Department of Molecular Microbiology, Washington University School of Medicine in St. Louis, St. Louis, MO, USA. [16]Department of Neurosurgery, Yale University School of Medicine, New Haven, CT, USA. [17]Yale Comprehensive Cancer Center, Yale University School of Medicine, New Haven, CT, USA. [18]Yale Stem Cell Center, Yale University School of Medicine, New Haven, CT, USA. [19]Yale Center for Biomedical Data Science, Yale University School of Medicine, New Haven, CT, USA. [20]These authors contributed equally: Lei Peng, Yingxia Hu, Madeleine C. Mankowski, Ping Ren. [21]These authors jointly supervised this work: Craig B. Wilen, Yong Xiong, Sidi Chen. ✉email: craig.wilen@yale.edu; yong.xiong@yale.edu; sidi.chen@yale.edu

