## [Peer Review File · Nature Communications]

Reviewer comments, initial review

Reviewer #1 (Remarks to the Author):

Peng et al immunized two common mouse model systems with SARS-CoV-2 RBD to generate neutralizing antibodies. The study reported two neutralizing antibodies clone2 and clone6 that showed high binding affinity to RBD and high pseudovirus neutralization potency. The determined Cryo-EM structures showed that clone2 and clone6 have similar RBD binding modes which are unique compared to published human SARS-CoV-2 neutralizing antibodies. The two antibodies also neutralize the variant of concern B.1.617. The transgenic mice challenge experiment showed that clone2 and clone6 can protect animals in both prophylactic and therapeutic settings.

As SARS-COV-2 continue to evolve, the ongoing COVID-19 pandemic enters a new stage which vaccine breakthroughs are frequently reported. This asks for more therapeutic options that can treat the variants of concern such as the delta variant. This study characterized two mouse neutralizing antibodies with a unique binding mode and VOC tolerance that maybe potential leads for therapeutic development. Despite so, the study emphasized too much on the pseudovirus rather than the authentic live virus neutralization potency. The live virus neutralization data indicates that the potencies of both antibodies against the wildtype strain are not impressive compared to many therapeutic antibodies like 10933 and 10987 and nanobodies. The authentic virus neutralization as well as challenge studies against the VOC are crucial for demonstrating the value of the antibodies but are lacking. The language and clarity of the manuscript has to be improved.

Major concerns:

It is known that the backbone of the pseudovirus and the cell systems used are critical for the SARS-CoV-2 neutralization assay. Further controls, like the therapeutic antibodies 10933 and 10987, are important for both pseudovirus and authentic virus neutralization assays.

Readability has to be improved at many places (page 5 line 4-5, page 8 line 21, page 19 line 30, Figure 4 legend, etc.).

It is redundant to show repeats of the pseudovirus neutralization data of clone2 and clone6 in Figure 4 a-f.

The key atom level interactions involved in epitope recognition were not shown in detail.

Other frequent RBD mutations in circulating SARS-CoV-2 strains like T478K and S477N are also close to the binding sites of clone2 and clone6, a thorough understanding of resistant mutations will be helpful for therapeutics.

The current 'humanization' of clone 2 and clone 6 did not change the framework regions of the variable domains to those of human antibodies, it is unclear whether that will be a challenge.

The method for clone 13A humanization and engineering was not described.

Minor issues:

Figure 3a and 3f have low resolutions.

Extended figure 2a, 2b, 2c, IC50s were not listed.

Reviewer #2 (Remarks to the Author):

Here, Dr. Peng and colleagues combined SARS-CoV-2 Spike RBD protein immunization with high-throughput single cell BCR sequencing technology to establish a platform to rapidly develop neutralizing antibody candidates and identified two neutralizing mAbs and generated a bispecific antibody of these two lead clones. The two monospecific and/or bispecific antibodies showed strong neutralization ability against SARS-CoV-2 and Indian variant lineage B.1.617. The author also analyzed the structure of Clone 2 and Clone 6 to SARS-CoV-2 RBD by cryo-EM and solved cryo-EM structures at ~3Å resolution. The structure of clone 2 and clone 6 complex with SARS-CoV-2 Spike conformations distinct from existing antibodies.

Overall, the authors established the platform and tried to isolate neutralizing mAbs from the immunized mice and modify these potential mAbs. A major conclusion from the paper is that monospecific and bispecific mAbs from the lead two mAbs are resistant to currently circulating SARS-CoV-2 variants. But many studies have been reported about SARS-CoV-2 neutralizing antibodies isolated from convalescent individuals, and many of them exhibited potent neutralizing potency against the SARS-CoV-2 WT and new variants.

Major:

1. Why sorted total plasma B cell for NGS single cell BCR-seq instead of RBD-specific memory B cells?
2. Why choose clone frequency as the screen target? The ratio of somatic hypermutation can be considered, and there is no analysis of the somatic hypermutation of the chosen mAbs in the manuscript.
4. Whether the bi-specific purified by SEC, make it clear in the methods.
5. Add a figure of SDS-PAGE for the bi-specific mAbs.
6. Humanization of mouse mAbs is the basic requirement for clinic therapeutic mAbs candidates, the neutralization test and affinity test should include the humanized mAb (Clone 13A).
7. What is the superiority of the bi-specific mAb than Clone 2 and Clone 6?
8. The neutralization test of humanized mAbs to SARS-CoV-2 and B.1.671 in vivo should be included.
9. In the cryo-EM structure analysis, the authors described that the RBD-binding modes of the lead mAbs were different from all other SARS-CoV-2 neutralizing antibodies reported to date (page 7, Line 25-26). Can the authors comment on whether there is any neutralization advantage by binding as this conformation? Aside from blocking ACE2, would the authors expect that these antibodies would be more likely to trap S protein in prefusion state, inhibiting the formation of syncytia, or cell fusion mentioned in the manuscript? A comment about the relation between the structure and the inhibition of cell fusion could be added in the discussion as mentioned in this study (PMID: 33974910).
10. A bispecific antibody was generated using the antigen-specific variable regions of the 2 lead mAbs. Although this bispecific mAb exhibited similar neutralizing potency against WT and B.1.617 pseudoviruses as these 2 lead mAbs, there was relatively reduced efficacy against the B.1.351 pseudovirus. The authors would address the advances of the bispecific antibody in discussion for better understanding the meaning of such design.

Minor

- 1) Since antibodies have been purified, the concentration of antibodies should be used for the x-axis.

(Extended Data Figure 1).

2) The author described that they established a repaid platform for screening and identifying neutralizing Abs. A comment on the advantages of this technology platform, such as time- or labor-saving, would be addressed in the discussion.

3) If did these antibodies cross-neutralize other coronaviruses such as SARS-CoV or not?

4) The figure mentioned at Page 5, line 15-16 was Extended Data Fig. 2e-f, but there was no such figure named Extended Data Fig. 2f in the manuscript. Besides, there is mistake at Page 6, line 20. The manuscript need to be checked carefully.

Reviewer #3 (Remarks to the Author):

The manuscript from Peng, Hu, Mankowski et al reports the isolation and characterization of SARS-CoV-2 spike RBD-directed antibodies from mice immunized with recombinant RBD. Two lead antibodies, Clone 2 and Clone 6, were chosen for further characterization, and these were demonstrated to bind with ~2 nM affinity to the RBD. Cryo-EM structures of these two antibodies bound to trimeric spike proteins revealed the RBD conformations compatible with antibody binding as well as the antibody epitopes. The two antibodies bind similarly, and their binding modes are also similar to other previously described antibodies, although there are some differences in angles of approach. The antibodies neutralize the wildtype and Delta variant equally well, but have 1-2 logs reduced potency against the Beta variant. Both antibodies are able to protect mice from severe disease caused by wildtype SARS-CoV-2 challenge when administered 24 hours before or 18 hours after challenge.

In general, the experiments are performed well and the conclusions are supported by the data. This is a thorough characterization of these antibodies. One weakness is the extent to which these data represent an advance over similar published studies. There is no shortage of antibody isolation and characterization papers for SARS-CoV-2. Most of the published papers characterize antibodies isolated from humans, whereas the antibodies described here were isolated from mice. For ease of development, human antibodies are preferred to humanized murine antibodies. Another weakness is that the antibodies described here are not broadly neutralizing, as evidenced by the reduced potency against the B.1351 (beta/SA) variant. There are now many antibodies described that show little to no reduction in potency against all variants of concern. The antibodies described here may be of some value, but not as much as other recently described human-derived antibodies.

Comments

1. It is very difficult to assess the quality of the EM maps, particularly at the antibody interfaces. The overall resolution is fine, but resolution in EM maps varies, and the authors do not show local resolution estimates. These must be shown. Zoomed views of the maps at the interface should also be shown. For ED Figure 3, the authors should just color the maps, rather than making them transparent with the ribbons underneath. Most cryo-EM studies of spike-antibody complexes perform focused/local refinement to improve the resolution of the map at the antibody interface, yet the authors do not perform these refinements.

2. The structure panels in 4J are not clear at all. Additional attention should be paid to these panels

to make the points clear to a general audience.

3. In general, the figures are very large. Some trimming is required. Figures 1a and 1b can be moved to supplemental or omitted completely.

4. The BLI fits need to be shown in Figure 1f.

5. The discussion is not well written. It is just a summary of the results. The authors should use this section to place their findings into context of the published SARS-CoV-2 antibody literature, including pre-prints. The readers should know how these antibodies compare to others in terms of potency, breadth, and binding modes.

Response to review (Peng et al. Nature Communications)

We thank the reviewers for their constructive comments. We have since performed a series of experiments, gather substantial new data, and fully revised the manuscript.

Summary of revision:

- Performed authentic virus of lead clones including humanized Clone 13A vs. Delta, in parallel with control antibodies (BL3), showing potent neutralization activity
- Performed in vivo challenge of humanized Clone 13A against authentic virus (BL3), both the original Wuhan/WA1 and the Delta variant, showing 100% protection efficacy in vivo
- Performed purification and biophysical characterization of humanized Clone 13A
- Performed additional 3D structural analysis
- Perform various other analyses to address all remaining critiques
- Including additional interpretations and discussions
- Fully revised the manuscript.

We provide point-by-point response to each comment below.

REVIEWER COMMENTS

Reviewer #1 (Remarks to the Author):

Peng et al immunized two common mouse model systems with SARS-CoV-2 RBD to generate neutralizing antibodies. The study reported two neutralizing antibodies clone2 and clone6 that showed high binding affinity to RBD and high pseudovirus neutralization potency. The determined Cryo-EM structures showed that clone2 and clone6 have similar RBD binding modes which are unique compared to published human SARS-CoV-2 neutralizing antibodies. The two antibodies also neutralize the variant of concern B.1.617. The transgenic mice challenge experiment showed that clone2 and clone6 can protect animals in both prophylactic and therapeutic settings.

As SARS-COV-2 continue to evolve, the ongoing COVID-19 pandemic enters a new stage which vaccine breakthroughs are frequently reported. This asks for more therapeutic options that can treat the variants of concern such as the delta variant. This study characterized two mouse neutralizing antibodies with a unique binding mode and VOC tolerance that maybe potential leads for therapeutic development. Despite so, the study emphasized too much on the pseudovirus rather than the authentic live virus neutralization potency. The live virus neutralization data indicates that the potencies of both antibodies against the wildtype strain are not impressive compared to many therapeutic antibodies like 10933 and 10987 and nanobodies. The authentic virus neutralization as well as challenge studies against the VOC are crucial for demonstrating the value of the antibodies but are lacking. The language and clarity of the manuscript has to be improved.

Response:

We thank the reviewer for the overall comments. In the revised manuscript, we added substantial authentic virus data, both in vitro and in vivo. We address the specific comments below.

Major concerns:

It is known that the backbone of the pseudovirus and the cell systems used are critical for the SARS-CoV-2 neutralization assay. Further controls, like the therapeutic antibodies 10933 and 10987, are important for both pseudovirus and authentic virus neutralization assays.

Response:

We added authentic virus data, both in vitro and in vivo. (**See new Figure 6**)

In an authentic virus neutralization assay against both WA and B.1.617.2 viruses, we tested our own clones alongside with control therapeutic antibodies RGEN 10933 and 10987. Our data showed that our antibodies neutralized authentic SARS-

CoV-2 virus in a potent manner. Potency drops accordingly, relatively little for RGEN 10933, Clone 2 and Clone 13A; moderate for Clone 6, and severe for RGEN 19087.

We have also added in vivo data of our humanized clone (Clone 13A) against authentic virus, both the original WA1 and Delta challenges. Results showed that the humanized clone showed full (100%) protection against both WA1 and Delta challenges.

Readability has to be improved at many places (page 5 line 4-5, page 8 line 21, page 19 line 30, Figure 4 legend, etc.).

Response:

In the revised manuscript, we improved the readability of these places.

It is redundant to show repeats of the pseudovirus neutralization data of clone2 and clone6 in Figure 4 a-f.

Response:

These neutralization data are different experiments, one set (a-c) to test a bi-specific clone (clone 16), the other set (e-f) to test combination (as a cocktail) of the two clones (Combo = Clone 2 + Clone 6). In both sets, we used single Clone 2 or

Clone 6 in parallel. These are two different experiments that are not redundant. We apologize for the confusion and further clarified this.

The key atom level interactions involved in epitope recognition were not shown in detail.

Response:

Thanks for the comments. We have included the detailed atomic interactions involved in the RBD-Fab interfaces in Extended Data Fig. 5 and included a discussion in the main text.

Extended Data Figure 5

Other frequent RBD mutations in circulating SARS-CoV-2 strains like T478K and S477N are also close to the binding sites of clone2 and clone6, a thorough understanding of resistant mutations will be helpful for therapeutics.

Response:

Thanks for the suggestion. We have redrawn Fig. 4j (Now Fig. 4g) and included all RBD mutations in the SA and Indian variants, along with another suggested frequent RBD mutation S477N in circulating SARS-CoV-2 strains. We have also rewritten the related result and discussion sections in the main text and analyzed the structural basis of potential impacts of these RBD mutations on Clone 2 and Clone 6 recognitions.

g

The current ‘humanization’ of clone 2 and clone 6 did not change the framework regions of the variable domains to those of human antibodies, it is unclear whether that will be a challenge.

The method for clone 13A humanization and engineering was not described.

Response:

We appreciate the comments. The humanization method we took is a standard approach in the field. We provided methods for this as well in the revised manuscript.

In our added data on biophysical characterization, neutralization assay against pseudovirus and authentic virus, and in vivo mouse challenge experiments against WA1 and Delta variant of SARS-CoV-2, clone 13A performed very well, suggesting that it is clear this won’t be a challenge at least in terms of the biophysical properties and functionality of this clone.

Minor issues:

Figure 3a and 3f have low resolutions.

Response:

Thanks for pointing this out. We have replaced the figures with high resolution images and revised Figure 3.

Extended figure 2a, 2b, 2c, IC50s were not listed.

Response:

For ext. Fig.2a-c, noted that these are preliminary clone screening experiments to find top clones, thus they are supplementary. The clone IC50 measurements were repeated, and the key IC50 data of the lead clones are in the main figures.

Reviewer #2 (Remarks to the Author):

Here, Dr.Peng and colleagues combined SARS-CoV-2 Spike RBD protein immunization with high-throughput single cell BCR sequencing technology to establish a platform to rapidly develop neutralizing antibody candidates and identified two neutralizing mAbs and generated a bispecific antibody of these two lead clones. The two monospecific and/or bispecific antibodies showed strong neutralization ability against SARS-CoV-2 and Indian variant lineage B.1.617. The author also analyzed the structure of Clone 2 and Clone 6 to SARS-CoV-2 RBD by cryo-EM and solved cryo-EM structures at ~3Å resolution. The structure of clone 2 and clone 6 complex with SARS-CoV-2 Spike conformations distinct from existing antibodies.

Overall, the authors established the platform and tried to isolate neutralizing mAbs from the immunized mice and modify these potential mAbs. A major conclusion from the paper is that monospecific and bispecific mAbs from the lead two mAbs are resistant to currently circulating SARS-CoV-2 variants. But many studies have been reported about SARS-CoV-2 neutralizing antibodies isolated from convalescent individuals, and many of them exhibited potent neutralizing potency against the SARS-CoV-2 WT and new variants.

Response:

We thank the reviewer for the overall comments. We acknowledged that multiple COVID antibody papers were published. Nevertheless, here we provide entirely new clones, as well as data on their biophysical characterization, functionality against variants of concerns, in vivo efficacy as well as new CryoEM structures. We believe this level of substantial data is suitable to be shared with the field via the platform of Nature Communications.

We also diligently cited a number of representative published COVID mAb papers from the field.

Major:

1. Why sorted total plasma B cell for NGS single cell BCR-seq instead of RBD-specific memory B cells?
2. Why choose clone frequency as the screen target? The ratio of somatic hypermutation can be considered, and there is no analysis of the somatic hypermutation of the choosed mAbs in the manuscript.

Response:

RBD-specific sorting is another approach, however, flow sorting usually leads to substantial loss of viable cells. We therefore use the approach of total plasma B cells for NGS single cell CR-seq. Generally speaking, the antigen-specific B cells will be significantly amplified upon immunization, which is the basis of immunization-based antibody development. Somatic hypermutation is a natural process, which happens frequently in the expanded antigen-specific B cells. Our goal is to identify lead antibody clones against SARS-CoV-2, thus the approach to identify them is not the important topic of this study.

4. Whether the bi-specific purified by SEC, make it clear in the methods.

Response:

We made this purification approach clear.

5. Add a figure of SDS-PAGE for the bi-specific mAbs.

Response:

We added a figure of SDS-PAGE.

b

6. Humanization of mouse mAbs is the basic requirement for clinic therapeutic mAbs candidates, the neutralization test and affinity test should include the humanized mAb (Clone 13A).

Response:

We performed neutralization test and affinity measurement experiments, and added a new data figure focusing on the humanized clone (Clone 13A). (See new Figure 6)

Figure 6

7. What is the superiority of the bi-specific mAb than Clone 2 and Clone 6?

Response:

The advantage of a bi-specific mAb is that it can recognize two epitopes. In the case of COVID-19, SARS-CoV-2 frequently evolves new mutations on Spike, and thereby escape from antibody recognition. Having a bi-specific can reduce the possibility of escaping both epitopes. This can also be achieved by mixing of two monoclonals, however two molecules will need to be made instead of one in that case, doubling the cost and creating potential combination or formulation complexity. The neutralization against WT spike is similar between the bi-specific and the monoclonals though.

8. The neutralization test of humanized mAbs to SARS-CoV-2 and B.1.671 in vivo should be included.

Response:

We performed additional experiments on neutralization of the humanized clone (Clone 13A), and added authentic virus data, both in vitro and in vivo. In an authentic virus neutralization assay against both WA and B.1.617.2 viruses, we tested our own clones alongside with control therapeutic antibodies RGEN 10933 and 10987. We have also added in vivo data of our humanized clone against authentic SARS-CoV-2 virus, both WA1 and Delta (the currently dominant variant). (**see new Figure 6) (same as above).**

9. In the cryo-EM structure analysis, the authors described that the RBD-binding modes of the lead mAbs were different from all other SARS-CoV-2 neutralizing antibodies reported to date (page 7, Line 25-26). Can the authors comment on whether there is any neutralization advantage by binding as this conformation? Aside from blocking ACE2, would the authors expect that these antibodies would be more likely to trap S protein in prefusion state, inhibiting the formation of syncytia, or cell fusion mentioned in the manuscript? A comment about the relation between the structure and the inhibition of cell fusion could be added in the discussion as mentioned in this study (PMID: 33974910).

Response:

Thanks for the suggestion. We have included a discussion in the main text along with an Extended Data Fig. 4e to analyze the relation between our antibody-bound S conformations and cell fusion. We also cited the study the reviewer suggested.

10. A bispecific antibody was generated using the antigen-specific variable regions of the 2 lead mAbs. Although this bispecific mAb exhibited similar neutralizing potency against WT and B.1.617 pseudoviruses as these 2 lead mAbs, there was relatively reduced efficacy against the B.1.351 pseudovirus. The authors would address the advances of the bispecific antibody in discussion for better understanding the meaning of such design.

Response:

The advantage of a bi-specific mAb is that it can recognize two epitopes. In the case of COVID-19, SARS-CoV-2 frequently evolves new mutations on Spike, and thereby escape from antibody recognition. Having a bi-specific can reduce the possibility of escaping both epitopes. This can also be achieved by mixing of two monoclonals, however two molecules will need to be made instead of one in that case. We also noted that the neutralization of the bi-specific antibody (Clone 16) has its own limitations. We discuss these points in the revised manuscript.

Minor

1) Since antibodies have been purified, the concentration of antibodies should be used for the x-axis. (Extended Data Figure 1).

Response:

In Extended Data Fig. 1e, we provided the antibody concentrations in the revised figure. In Extended Data Figure 1c, these are ELISAs for initial clone screening experiments, where only the dilution is standardized, thus the concentration is not applicable.

2) The author described that they established a repaid platform for screening and identifying neutralizing Abs. A comment on the advantages of this technology platform, such as time- or labor-saving, would be addressed in the discussion.

Response:

Thanks for the suggestion. We have discussed the advantages of this platform, for example, its simplicity and human patient sample independence (therefore reduced biohazard of potential infectious agents). Also, antibodies isolated from immunized mice sometimes can have higher affinity than human antibodies from natural infection, because immunization of mice can use pure, high dose antigens.

3) If did these antibodies cross-neutralize other coronaviruses such as SARS-CoV or not?

Response:

Thanks for the suggestion. We have performed the neutralization assays and found that these mAbs do not cross-neutralize SARS-CoV pseudovirus.

4) The figure mentioned at Page 5, line 15-16 was Extended Data Fig. 2e-f, but there was no such figure named Extended Data Fig. 2f in the manuscript. Besides, there is mistake at Page 6, line 20. The manuscript need to be checked carefully.

Response:

We apologize for these typos. We have fixed these issues in the revised manuscript.

Reviewer #3 (Remarks to the Author):

The manuscript from Peng, Hu, Mankowski et al reports the isolation and characterization of SARS-CoV-2 spike RBD-directed antibodies from mice immunized with recombinant RBD. Two lead antibodies, Clone 2 and Clone 6, were chosen for further characterization, and these were demonstrated to bind with ~2 nM affinity to the RBD. Cryo-EM structures of these two antibodies bound to trimeric spike proteins revealed the RBD conformations compatible with antibody binding as well as the antibody epitopes. The two antibodies bind similarly, and their binding modes are also similar to other previously described antibodies, although there are some differences in angles of approach. The antibodies neutralize the wildtype and Delta variant equally well, but have 1-2 logs reduced potency against the Beta variant. Both antibodies are able to protect mice from severe disease caused by wildtype SARS-CoV-2 challenge when administered 24 hours before or 18 hours after challenge.

In general, the experiments are performed well and the conclusions are supported by the data. This is a thorough characterization of these antibodies. One weakness is the extent to which these data represent an advance over similar published studies. There is no shortage of antibody isolation and characterization papers for SARS-CoV-2. Most of the published papers characterize antibodies isolated from humans, whereas the antibodies described here were isolated from mice. For ease of development, human antibodies are preferred to humanized murine antibodies. Another weakness is that the antibodies described here are not broadly neutralizing, as evidenced by the reduced potency against the B.1351 (beta/SA) variant. There are now many antibodies described that show little to no reduction in potency against all variants of concern. The antibodies described here may be of some value, but not as much as other recently described human-derived antibodies.

Response:

We thank the reviewer for the comments. We acknowledged that multiple COVID antibodies exist at this point. Nevertheless, here we provide entirely new clones, as well as data on their biophysical characterization, functionality against variants of concerns, in vivo efficacy as well as new CryoEM structures. We believe these data is suitable to be shared with the field.

Comments

1. It is very difficult to assess the quality of the EM maps, particularly at the antibody interfaces. The overall resolution is fine, but resolution in EM maps varies, and the authors do not show local resolution estimates. These must be shown. Zoomed views of the maps at the interface should also be shown. For ED Figure 3, the authors should just color the maps, rather than making them transparent with the ribbons underneath. Most cryo-EM studies of spike-antibody complexes perform focused/local refinement to improve the resolution of the map at the antibody interface, yet the authors do not perform these refinements.

Response:

Thanks for the suggestion, we have included the local resolution estimations for the complex structures (new Extended Data Fig. 7).

Zoomed in views of the maps at the Fab6-RBD and Fab2-RBD interfaces are also shown (new Extended Data Fig. 5).

For ED Fig.3, we have replaced with colored maps without the ribbon models. Perhaps we did not describe clearly, we did perform multibody refinement by dividing the spike-Fab complex into 4 bodies (3 RBD-Fab protomers, and the spike trimer core) for the spike-Fab6 complex, as well as masked refinement focusing on each RBD-Fab protomer followed by multibody refinement for the spike-Fab2 complex, which has been described in details in the method section.

Extended Data Figure 3

2. The structure panels in 4J are not clear at all. Additional attention should be paid to these panels to make the points clear to a general audience.

Response:

We have updated Fig. 4J (now Fig. 4g) with detailed RBD-Fab interactions at the interface of WT spike and variants, and have rewritten in the main text the structural basis of the impact of these mutations on the antibody neutralization potencies to the SARS-CoV-2 spike.

g

3. In general, the figures are very large. Some trimming is required. Figures 1a and 1b can be moved to supplemental or omitted completely.

Response:

We have revised Figure 1 and moved 1a/1b to supplement.

4. The BLI fits need to be shown in Figure 1f.

Response:

We have provided updated Figure 1f (now Fig. 1c) to show BLI fits.

Of note, the fitting curves for clone6 at high concentrations were missing because they were not included in the fitting. This is because the binding is particularly strong due to high avidity, dissociation never occurred, yielding the K_d at low picomolar. The binding is too strong and the BLI sensors were not saturated before dipping into the buffer. This was observed before: In the Cell paper (<https://doi.org/10.1016/j.cell.2020.05.025>) Figure S5A, one Ab BD-218 with particularly strong binding also has high-concentration fitting curve missing.

5. The discussion is not well written. It is just a summary of the results. The authors should use this section to place their findings into context of the published SARS-CoV-2 antibody literature, including pre-prints. The readers should know how these antibodies compare to others in terms of potency, breadth, and binding modes.

Response:

We acknowledged that multiple COVID antibodies exist at this point. Since there is a breath of literature on this topic, we have cited a number of existing antibodies reported in primary literature and preprints. We also apologize for being unable to cite all COVID antibody papers in limited space. We revised the discussion of the manuscript accordingly.

Reviewer comments, second review

Reviewer #1 (Remarks to the Author):

Overall, the revised manuscript improves substantially. I have no more major concerns.

Minors:

Figure S5, it is better label polar interactions.

Figure 4g has a low resolution.

The manuscript has the following typos:

P12I7, 'B.1.1351' should be B.1.351.

P12I20, 'RGEN 19087' should be 'REGN-10987'.

µg and ug should be unified in text and figures.

Reviewer #2 (Remarks to the Author):

In this revision of "Monospecific and bispecific monoclonal SARS-CoV-2 neutralizing antibodies that maintain potency against B.1.617", the authors have thoughtfully and thoroughly responded to the concerns and questions raised by this reviewer. Collectively, these data and alterations of the text have strengthened the manuscript and also improved its clarity.

Reviewer #3 (Remarks to the Author):

The authors have substantially revised and improved their manuscript. The quality of the EM maps is now easier to ascertain, as are the BLI fits. The discussion is also improved and the manuscript is suitable for publication.

Response to review (Peng et al. Nature Communications)

We thank the reviewers for their constructive comments. We provide point-by-point response to each comment below.

REVIEWERS' COMMENTS

Reviewer #1 (Remarks to the Author):

Overall, the revised manuscript improves substantially. I have no more major concerns.

Minors:

Figure S5, it is better label polar interactions.

Figure 4g has a low resolution.

Response:

In the revised manuscript, we have relabeled the polar interaction in Figure S5 and replaced Figure 4g with a higher low resolution.

The manuscript has the following typos:

P12I7, 'B.1.1351' should be B.1.351.

P12I20, 'RGEN 19087' should be 'REGN-10987'.

□g and ug should be unified in text and figures.

Response:

In the revised manuscript, we corrected B.1.1351 into B.1.351. RGEN 19087 into REGN-10987. We unified the μg in text and figures.

Reviewer #2 (Remarks to the Author):

In this revision of "Monospecific and bispecific monoclonal SARS-CoV-2 neutralizing antibodies that maintain potency against B.1.617", the authors have thoughtfully and thoroughly responded to the concerns and questions raised by this reviewer. Collectively, these data and alterations of the text have strengthened the manuscript and also improved its clarity.

Response:

We thank the reviewer again for reading and approving the revision.

Reviewer #3 (Remarks to the Author):

The authors have substantially revised and improved their manuscript. The quality of the EM maps is now easier to ascertain, as are the BLI fits. The discussion is also improved and the manuscript is suitable for publication.

Response:

We thank the reviewer again for reading and approving the revision.